# DNA double-strand break end synapsis by DNA loop extrusion

Jin H. Yang [1,2,3], Hugo B. Brandão [1,2,3,4] ✉ & Anders S. Hansen [1,2,3] ✉

DNA double-strand breaks (DSBs) occur every cell cycle and must be efficiently repaired. Non-homologous end joining (NHEJ) is the dominant pathway for DSB repair in G1-phase. The first step of NHEJ is to bring the two DSB ends back into proximity (synapsis). Although synapsis is generally assumed to occur through passive diffusion, we show that passive diffusion is unlikely to produce the synapsis speed observed in cells. Instead, we hypothesize that DNA loop extrusion facilitates synapsis. By combining experimentally constrained simulations and theory, we show that a simple loop extrusion model constrained by previous live-cell imaging data only modestly accelerates synapsis. Instead, an expanded loop extrusion model with targeted loading of loop extruding factors (LEFs), a small portion of long-lived LEFs, and LEF stabilization by boundary elements and DSB ends achieves fast synapsis with near 100% efficiency. We propose that loop extrusion contributes to DSB repair by mediating fast synapsis.

DNA double-strand breaks (DSBs) can be caused by environmental agents such as radiation and drugs[1–3] and endogenous metabolism such as transcription and replication stress[4,5]. For example, normal metabolism has been estimated to cause ~1–50 DSBs per human cell per day[6,7]. Consequently, fast and reliable DNA repair is necessary to prevent deleterious chromosomal rearrangements such as translocations, inversions, amplifications, and deletions[8]. The three major DSB repair pathways are non-homologous end-joining (NHEJ), alternative end-joining (Alt-EJ), and homologous recombination (HR)[9–11]. The choice of DSB pathway depends on sequence, chromatin context, cell cycle phase, and the complexity of DSB ends[9,11,12]. Here we focus on NHEJ, which is operational throughout the cell cycle and the dominant DSB repair pathway in G1-phase[13].

While much is known about the proteins that are recruited to DSB ends, their order of recruitment, and the molecular mechanisms involved in the repair[3,14,15], what all recruitment mechanisms have in common is that they are reactive: recruitment of DSB repair factors begins only after the DSB has occurred and been sensed. This introduces a time delay[16] during which the DSB ends can diffuse apart[17], which may delay the repair, prevent repair, or result in aberrant ligation between distinct chromosomes, causing translocations[18]. Indeed,

prior experimental work has demonstrated that, in human cells, DSB ends can move several hundreds of nanometers apart within minutes after a DSB has occurred[19,20]. The DNA DSB repair process through NHEJ, therefore, requires two major steps: (1) bringing the DSB ends back into proximity (this process is called synapsis[21]) and (2) recruiting the necessary proteins to covalently ligate the synapsed DSB ends (Fig. 1a). While much is known about the second NHEJ step, the alignment and covalent linkage of synapsed broken DNA ends[11,22], comparatively less is known about the first step, i.e., how the two DSB ends are brought into proximity to achieve synapsis.

Synapsis, the bringing of DSB ends back together for NHEJ, is generally assumed to be mediated by passive 3D diffusion[17,19,21,23–25] (Fig. 1b). However, here we show (see Results below) that passive diffusion is likely too slow to be consistent with DSB repair by NHEJ in G1-phase observed in mammalian cells[26]. The inability of passive diffusion to explain the kinetics of synapsis in vivo suggests that alternative mechanisms must be operational inside the cell. Here, we hypothesize that DNA loop extrusion contributes to DSB repair by mediating fast and efficient DSB end synapsis.

Loop extrusion is emerging as a universal mechanism that folds genomes into loops and domains[27]. In mammalian interphase, the

[1]Department of Biological Engineering, Massachusetts Institute of Technology, Cambridge, MA 02139, USA. [2]The Broad Institute of MIT and Harvard, Cambridge, MA 02142, USA. [3]Koch Institute for Integrative Cancer Research, Cambridge, MA 02142, USA. [4]Present address: Illumina Inc., San Diego, CA 92122, USA. ✉e-mail: articles@hugoresearch.com; ashansen@mit.edu

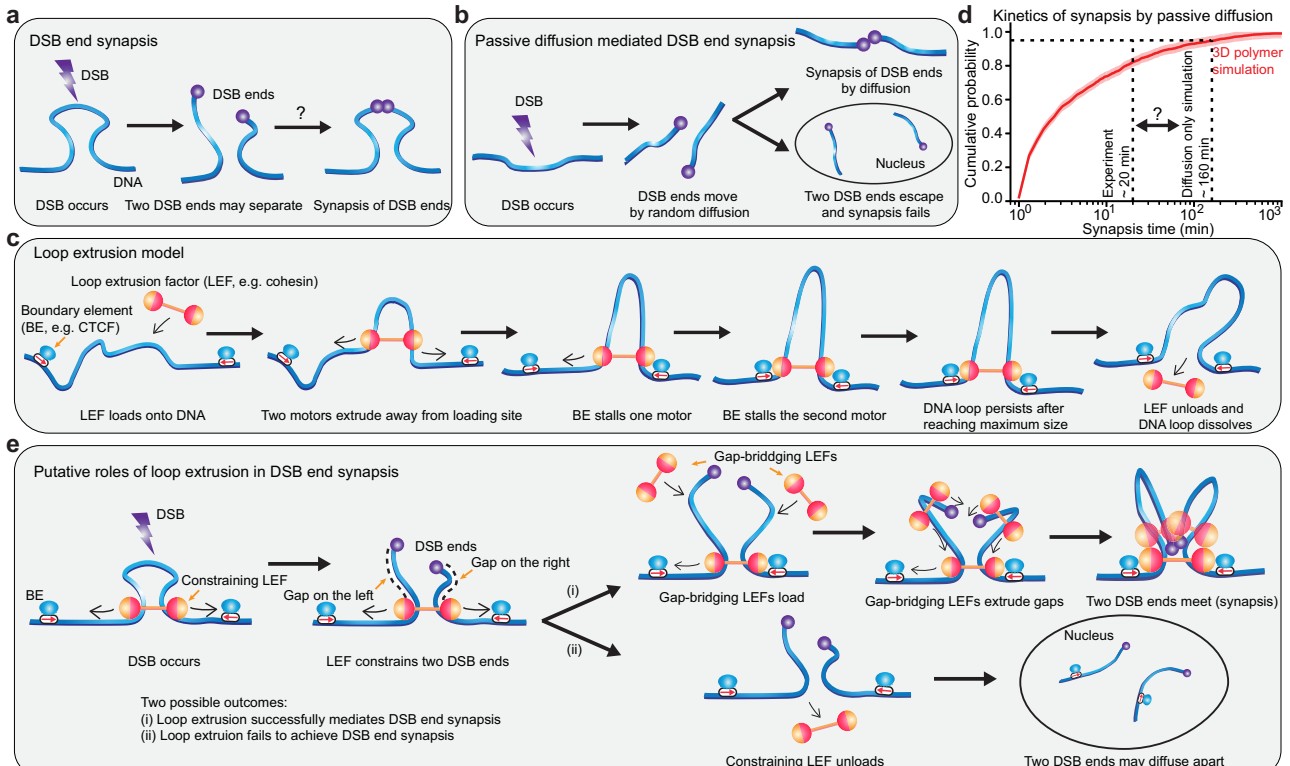

**Fig. 1 | A model of DSB synapsis, mediated by DNA loop extrusion. a** After a DSB has occurred, the two DSB ends may separate. How the two DSB ends are constrained from diffusing too far apart and brought back into proximity for downstream repair is not well understood. **b** Overview of DSB end synapsis mediated by passive diffusion. **c** Overview of the loop extrusion model. Loop extruding factors (LEFs) extrude bidirectionally away from the loading site; the two motors of a LEF extrude independently: after one motor is stalled by a boundary element (BE), the other motor can continue extruding until encountering a BE on the other side. **d** 3D polymer simulation reveals a large discrepancy between the kinetics of synapsis mediated by diffusion alone and the synapsis kinetics determined experimentally. The shaded area around the cumulative probability curve (calculated from 2223

DSB events) represents the 95% confidence interval of the cumulative probability estimated with Dvoretzky–Kiefer–Wolfowitz inequality. The vertical dash lines indicate the experimentally determined synapsis time to reach 95% and the simulated synapsis time by passive diffusion alone to reach 95%, respectively. **e** Loop extrusion may facilitate DSB end synapsis in two ways: (1) the constraining LEF prevents the two DSB ends from diffusing apart after DSB; (2) Additional gap-bridging LEFs loaded within the loop extruded by constraining LEF can extrude sub-loops to bring the two DSB ends into proximity. However, if the constraining LEF falls off before the two DSB ends are brought into proximity by gap-bridging LEFs, the two DSB ends may diffuse apart. In our simulations, we assume LEFs cannot pass one another or DSB ends.

primary loop extruding factor (LEF) is the cohesin complex, which is thought to extrude DNA bidirectionally at a rate of ~0.5–2 kb/s until cohesin is blocked by a Boundary Element (BE)[28–30]. The primary BE in mammalian interphase is the insulator protein, CTCF[31]. By extruding the genome until it encounters CTCF boundaries, cohesin-mediated loop extrusion folds the genome into loops and domains known as topologically associating domains (TADs)[32,33] (Fig. 1c). Beyond cohesin, several other structural maintenances of chromosomes (SMC) family complexes function as LEFs, including the condensin complexes that mediate mitotic chromosome compaction and bacterial SMC complexes that help resolve sister chromatids[27,34–37].

We propose that loop extrusion likely plays a role in DSB repair for at least three reasons. First, loop extrusion is operational across the entire interphase genome. Thus, the loop extrusion and DSB repair machinery will necessarily encounter each other when a DSB occurs and have to interact. Second, experimental estimates suggest that most interphase DNA is inside an extruding cohesin loop at any given time[38–41] (Supplementary Note 2.1). Thus, a DSB is more likely to occur inside a cohesin loop than outside. Third, unlike known reactive DSB repair mechanisms, loop extrusion could function as a preemptive mechanism that prevents the DSB ends from diffusing apart and simultaneously accelerate the synapsis process, thereby promoting fast and efficient DSB repair. To test this hypothesis, we combine analytical theory and polymer simulations, to quantitatively

investigate the extent to which loop extrusion may help DNA repair by facilitating synapsis for NHEJ. We find that DNA loop extrusion can promote very fast (~10 min) and efficient (≥95%) synapsis and identify the parameter regimes required for efficient synapsis. Finally, we make several experimentally testable predictions to probe the relationship between the role of loop extrusion and DSB synapsis in NHEJ.

## Results

### 3D diffusion may be too inefficient to promote DSB end synapsis in mammalian cells

We began by investigating the prevailing model: that passive 3D diffusion mediates DSB synapsis[17,19,21,23–25]. The most direct measurement of in vivo synapsis kinetics thus far shows ~95% of DSBs are synapsed within 20 min in human osteosarcoma (U2OS) cells[20]. Consistently, the average synapsis time is estimated to be around 6–17 min in mammalian cells based on prior experimental data[9,20,42,43]. To see if 3D diffusion-mediated synapsis is consistent with these values, we performed 3D polymer simulations to determine the kinetics of synapsis mediated by passive diffusion alone, using parameters from live-cell imaging experiments without cohesin and loop extrusion[41]. We found that passive diffusion takes ~160 min to achieve 95% DSB synapsis, almost an order of magnitude longer than the experimentally observed value of 20 min (Fig. 1d). This suggests the existence of alternative mechanisms that help accelerate DSB synapsis.

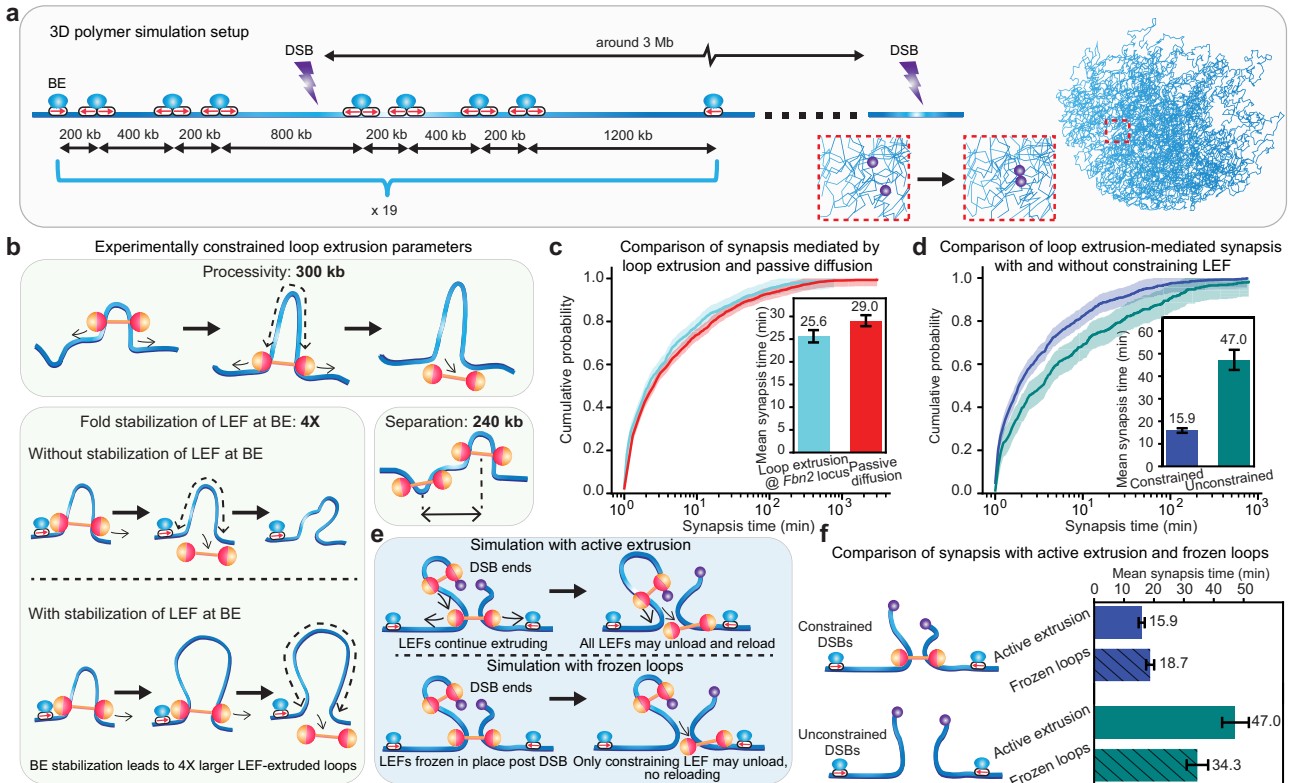

**Fig. 2 | Loop extrusion achieves faster synapsis than passive diffusion alone.**
**a** Overview of 3D polymer simulation setup and representative 3D polymer conformation. A snapshot of part of the chromosome used in 3D polymer simulations is shown on the right. The inset in the red dashed square depicts the synapsis of two DSB ends. **b** loop extrusion dynamic parameters estimated from the *Fbn2* locus. **c** Improved synapsis efficiency with loop extrusion compared with passive diffusion alone. Cumulative probabilities of synapsis time for synapsis with loop extrusion and with passive diffusion alone are calculated from 1376 and 2223 DSB events, respectively. **d** Higher synapsis efficiency at DSBs constrained by LEFs (941 events) than unconstrained DSBs (435 events). The shades in (**c, d**) around the cumulative probability curves represent the 95% confidence interval of the cumulative probability estimated with Dvoretzky–Kiefer–Wolfowitz inequality.
**e** Schematic diagram of 3D polymer simulations with active extrusion versus frozen

loop. In simulations with active extrusion, all LEFs may unload and reload prior to and after DSB occurrence. In simulations with frozen loops, all LEFs may unload and reload prior to DSB occurrence; after DSB occurrence, only constraining LEF may unload (but not reload), and all other LEFs are frozen in place. **f** Improved synapsis efficiency with active extrusion at constrained DSBs (blue bars) and reduced synapsis efficiency with active extrusion at unconstrained DSBs (green bars). In simulations with frozen loops, 750 DSBs were constrained and 357 DSBs were unconstrained. The error bars of the bar plot in (**c, d, f**) represent a 95% confidence interval of the mean using maximum likelihood estimation of the exponential distribution accounting for censored data[41]. Cumulative probabilities of synapsis time for synapsis at constrained and unconstrained DSBs in frozen loop situations are shown in Supplementary Fig. 8b.

## A plausible paradigm for DSB repair via DNA loop extrusion

Given the unphysiologically slow rates of DSB end synapsis mediated by passive diffusion alone, we hypothesize that DNA loop extrusion by loop extruding factors (LEFs) (such as cohesins, condensins, and other SMC complexes operational inside the nucleus[37]) facilitates DSB synapsis and repair in two ways, in parallel to 3D diffusion.

First, since most of the genome is inside LEF loops[38–40], a DSB is statistically more likely to occur inside a loop than outside. If a DSB occurs inside a LEF-mediated DNA loop, the DSB ends are constrained and unable to diffuse too far apart (Fig. 1e), and we call such a LEF a constraining LEF. The constraining LEFs' presence on DNA provides a time window of opportunity for the two DSB ends to synapse either through passive diffusion[17] or through the action of gap-bridging LEFs (explained below), where we define a gap as the DNA segment between the constraining LEF and the broken DNA end (Fig. 1e).

Second, while the constraining LEF holds together the two pieces of DNA, gap-bridging LEFs, dynamically loaded between one DSB end and the constraining LEF, can extrude loops that bring each DSB end into proximity with the constraining LEF (illustrated in Fig. 1e). If both sides of the DSB are extruded by gap-bridging LEFs, the DSB ends will be brought into spatial proximity, thereby achieving synapsis (Fig. 1e, top branch); the stochastic nature of LEFs binding to and unbinding

from gaps means that multiple attempts may be required to simultaneously bridge both gaps via gap-bridging LEFs before synapsis is achieved (Supplementary Fig. 1). For the models explored in this study, we assume LEFs cannot extrude past DSB ends (i.e., they do not "fall off" the DNA at the site of DSB), as supported by experimental evidence of DSB ends acting as cohesin roadblocks[44,45] and that LEFs do not bypass each other. Lastly, we note that any LEF can serve as "constraining" or "gap-bridging", and that designation only depends on its current position with respect to the DSB. For example, a constraining LEF unloaded from one locus can reload at another DSB site to function as a gap-bridging LEF.

We also note that loop extrusion will work in synergy with 3D diffusion. For example, if loop extrusion brings the DSB ends sufficiently close together, passive DNA diffusion, which works in parallel, might expedite the encounter of the two ends. We explore this synergy in the results that follow.

Thus, we set out to test our hypothesis that loop extrusion may contribute to DSB repair by facilitating synapsis by asking two questions: (1) Can the process of DNA loop extrusion accelerate synapsis? (2) If so, can we identify physiologically plausible conditions for loop extrusion (e.g. LEF density on DNA or LEF processivity) that achieve synapsis kinetics and efficiency observed in vivo?

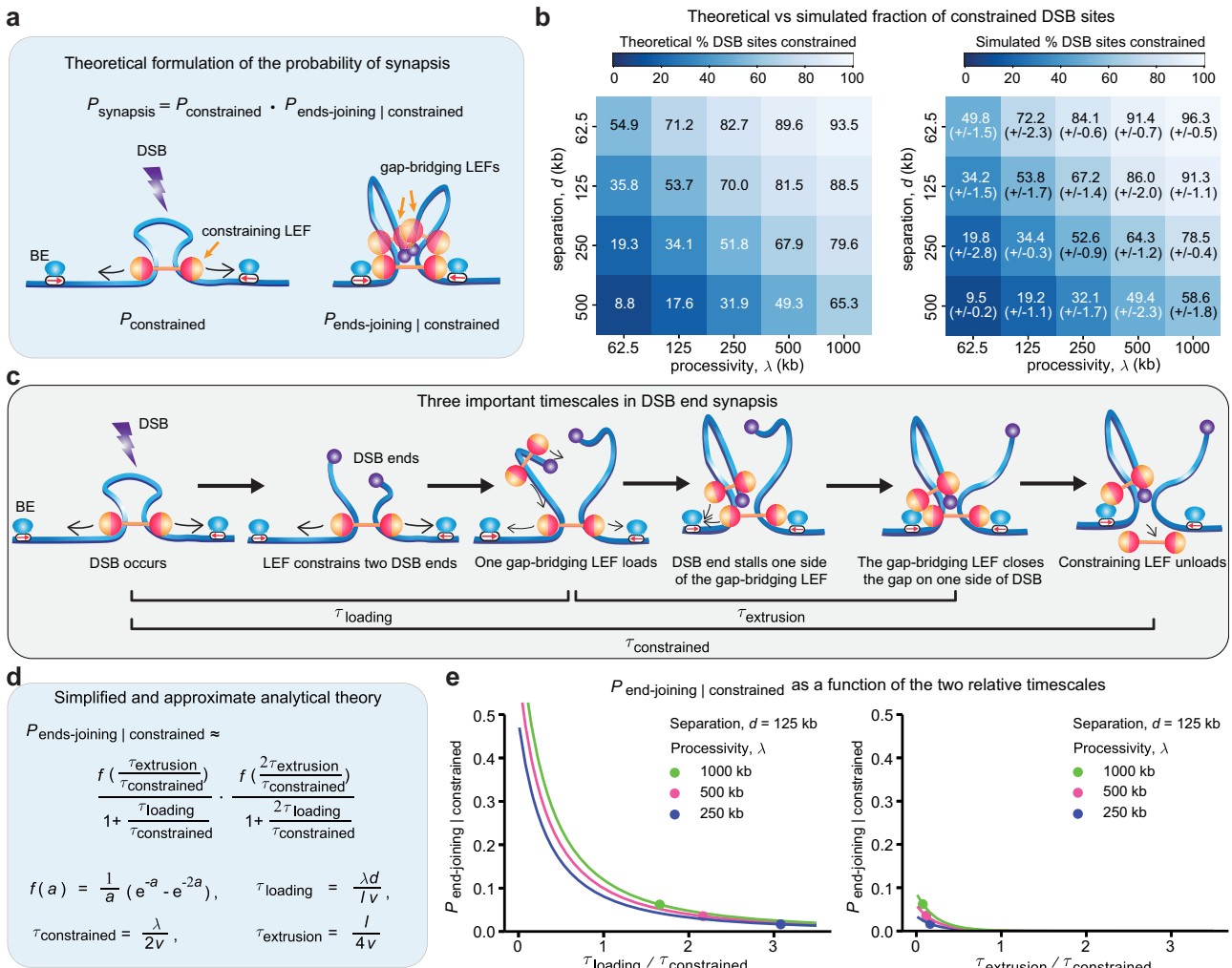

**Fig. 3 | Synapsis can be quantitatively predicted and mechanistically understood using an analytical theory. a** The probability of synapsis can be decomposed into the probability of being constrained and the conditional probability of gap-bridging given that the DSB was constrained. **b** $P_{constrained}$ can be predicted by the ratio of processivity and separation. Heatmaps of predicted (left) and simulated (right; numbers in brackets show standard error of the mean, $n = 3$ independent 1D simulations, with 216–218 DSB events per simulation) fraction of constrained DSB sites with different combinations of processivity (y-axis) and separation (x-axis). Boundary strength = 0.5 was used in the simulations. **c** Three important timescales in DSB end synapsis. **d** $P_{end\text{-joining}|constrained}$ is determined by two relative timescales: the ratio of loading time and constraining time, and the ratio of extrusion time and

constraining time. $\lambda$ is LEF processivity, $d$ is LEF separation, $l$ is the average LEF loop length (a function of $\lambda$ and $d$), and $v$ is the extrusion speed in one direction (i.e., 1/2 the total extrusion speed). **e** Larger improvement on synapsis efficiency can be achieved by reducing $\tau_{loading}/\tau_{constrained}$ than by reducing $\tau_{extrusion}/\tau_{constrained}$. The data points (circles) indicate the $P_{end\text{-joining}|constrained}$ at the separation and processivity indicated in the legend (no stabilization of LEFs at BE), whereas the line plots show how $P_{end\text{-joining}|constrained}$ varies with the ratio of loading time and constraining time (left panel) and the ratio of extrusion time and constraining time (right panel), while holding the other ratio constant at the values corresponding to the circle data points.

## Loop extrusion achieves faster synapsis than pure diffusion

To test our hypothesis that loop extrusion may promote DSB repair by facilitating DSB synapsis, we performed 3D polymer simulations with experimentally constrained loop extrusion dynamic parameters[41] that incorporate DSBs, LEFs, and boundary elements (BEs). We simulated one chromosome 70 Mb in length that could be bound by LEFs, where the chain was discretized into 1 kb DNA segments. BEs were placed on the chromosome to create TADs of various sizes within the experimentally expected range[46–48] (Fig. 2a). In our simulated chromosomal DNA, LEFs were allowed to dynamically load to any DNA segment not occupied by other LEFs, extrude loops, and dissociate. We introduced DSBs approximately every 3 Mb. We monitored the distance between each pair of DSB ends and recorded the events when the two DSB ends came within a pre-defined capture radius (see Fig. 2a and Methods). The simulation parameters for loop extrusion were inferred from experimental measurements of live-cell imaging and Micro-C data[41]

(Fig. 2b). We used: (1) LEF processivity of 300 kb, i.e., the average length of DNA extruded by an unobstructed LEF (processivity = LEF residence time × extrusion speed); (2) Average LEF separations of 240 kb (corresponding to ~4 LEFs per Mb); (3) A fourfold stabilization of LEF at BE, i.e., the fold increase of LEF processivity when LEF is stalled by BE. Finally, we used a BE boundary strength of 0.5, defined as the probability for a BE to stall LEF extrusion when a LEF encounters a BE (explored in Supplementary Fig. 2; synapsis efficiency is relatively insensitive to BE strength). We then compared the synapsis process simulated with the parameters described above with synapsis mediated by passive diffusion alone.

Analysis of our 3D polymer simulations shows that loop extrusion shifts the cumulative probability curve of synapsis time to the left (one-sided permutation test using 10,000 random permutations $P = 4.00 \times 10^{-4}$). The mean-synapsis time is modestly but significantly reduced compared with passive diffusion alone (Fig. 2c).

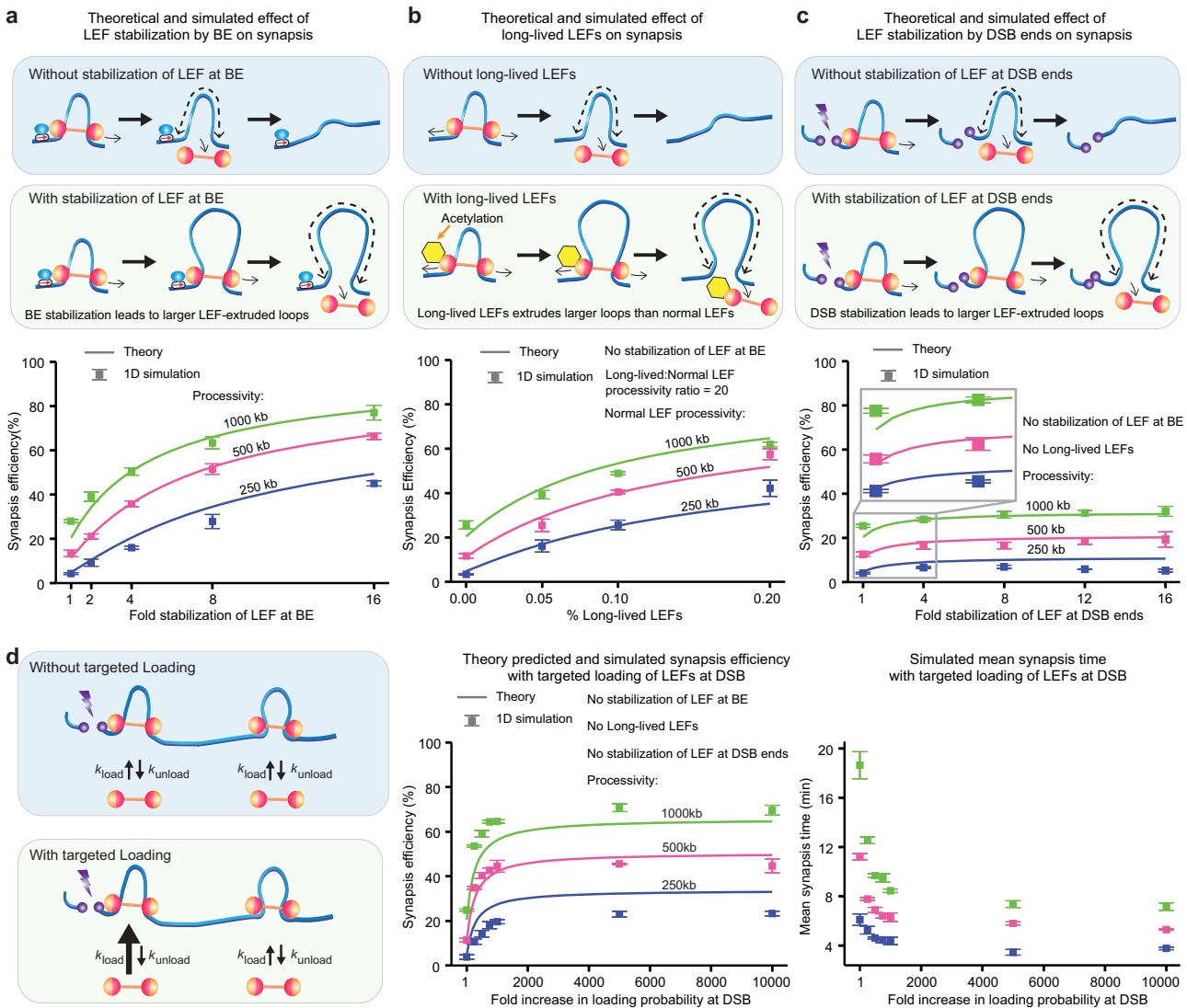

**Fig. 4 | LEF stabilization by either BEs or DSBs improves synapsis efficiency, as does the presence of long-lived LEFs and targeted loading of LEF at DSB. a–c** Schematic diagrams of the effects of stabilization of LEFs by BE, having a small portion of long-lived LEFs, and stabilization of LEFs by DSB (top), and the corresponding synapsis efficiency (bottom) predicted by theory (lines) or obtained from 1D simulations (squares; the error bars represent the standard error of the mean, $n = 3$ independent 1D simulations, with 216–218 DSB events per simulation). LEF separation of 125 kb and boundary strength of 0.5 were used. The inset in (**c**) shows the modest but statistically significant improvement in synapsis efficiency when

fold stabilization of LEF at DSB ends increases from 1 to 4. **d** Schematic diagrams of the effects of targeted loading of LEF at DSB (left), the corresponding synapsis efficiency (middle) predicted by theory (lines) or obtained from 1D simulations, and the corresponding mean-synapsis time (right). Conversion of 1D simulation time steps to synapsis time assumes a total extrusion speed of 1 kb/s. The error bars represent the standard error of the mean, $n = 3$ independent 1D simulations, with 216–218 DSB events per simulation. LEF separation of 125 kb and boundary strength of 0.5 were used.

We, therefore, investigated the underlying causes of the increased synapsis efficiency due to loop extrusion: we compared the synapsis process at DSBs with or without constraining LEFs. Consistent with our hypothesis, constrained DSBs have three times faster synapsis than unconstrained DSBs on average (Fig. 2d). To test the contribution of gap-bridging LEFs on synapsis efficiency, we performed a series of 3D polymer simulations with "frozen loops" (Fig. 2e), where LEFs are frozen in place post-DSB, and only constraining LEF may unload (but not reload), and compared it to the case where LEFs may continue to extrude and load/unload. We found that active extrusion significantly accelerates synapsis at constrained DSBs (two-sided $t$-test $P \leq 0.01$ from Kaplan–Meier corrected synapsis times). Surprisingly, however, for unconstrained DSBs, active extrusion was slightly detrimental to synapsis (Fig. 2f); one explanation is that without constraining LEFs, extrusion by gap-bridging LEFs may pull two DSB ends away from each

other and thereby slow down synapsis. These results emphasize the importance of constraining LEFs for efficient DSB repair.

In the simple loop extrusion model considered thus far, the overall synapsis kinetics still fails to match the experimentally observed kinetics. However, we find that in certain conditions, LEFs can help lower synapsis times by more than three-fold. Therefore, we sought to further dissect the loop extrusion-mediated synapsis process through analytical theory to better understand what factors can promote faster synapsis.

## Analytical theory elucidates two relative timescales underlying synapsis efficiency

We formulated an analytical theory that explicitly considers the sequential steps of LEF-mediated synapsis, and computes the probability of synapsis as a function of loop extrusion parameters

(Supplementary Note 1). We found that the probability of LEF-mediated synapsis, $P_{synapsis}$, can be decomposed into the product of two probabilities: the probability of the DSB occurring inside a DNA loop such that the broken DNA ends remain constrained by LEFs, $P_{constrained}$, and the conditional probability of end-joining given that the DSB is constrained, $P_{end\text{-}joining|constrained}$ (Fig. 3a and Supplementary Note 1.1).

We note that loop extrusion only facilitates synapsis when its constraining role is in effect (Fig. 2c–f). As such, $P_{constrained}$ sets an upper bound for the $P_{synapsis}$. Therefore, to maximize synapsis efficiency, it is necessary to have $P_{constrained}$ as close to 1 as possible. $P_{constrained}$ can be calculated as the fraction of the genome that is covered by LEF-mediated DNA loops. By accounting for the effect of BEs on LEFs, we derived an expression for $P_{constrained}$ that accurately estimates the fraction of the genome inside loops (see Eq. (27) in Supplementary Note 1.2), as shown in Fig. 3b. Given a large number of processivity-separation combinations, hereafter we fix the LEF separation at 125 kb unless otherwise specified, which gives the highest genome coverage by LEFs among separation values estimated by studies comparing polymer simulations and experimental data[32,41,49–54].

Next, to understand the role of gap-bridging LEFs in mediating synapsis, we examined how $P_{end\text{-}joining|constrained}$ modulates synapsis efficiency. We derived a general analytical expression for $P_{end\text{-}joining|constrained}$ that accounts for gap-bridging LEFs that load and finish extruding gaps on both sides of the DSB before the constraining LEF unloads. With simplifying assumptions (Eqs. (67–70), see Supplementary Note 1.3), we identified two relative timescales that dominate $P_{end\text{-}joining|constrained}$: $\tau_{loading}/\tau_{constrained}$ and $\tau_{extrusion}/\tau_{constrained}$ (Fig. 3c, d and Supplementary Note 1.3). $\tau_{loading}$ is the duration for gap-bridging LEFs to load into the gap between the DSB and the constraining LEF; $\tau_{constrained}$ is the time from DSB occurrence to unloading of the constraining LEF; $\tau_{extrusion}$ is the time for the gap-bridging LEFs to finish extruding the DNA between the DSB end and the constraining LEF. We find that while reducing either relative timescale improves synapsis efficiency, larger improvement can be achieved by reducing $\tau_{loading}/\tau_{constrained}$ than by reducing $\tau_{extrusion}/\tau_{constrained}$ (Fig. 3e). Indeed, a comparison of $\tau_{loading}/\tau_{constrained}$ and $\tau_{extrusion}/\tau_{constrained}$ across different combinations of LEF processivities and separations shows that the loading time of gap-bridging LEFs relative to the constraining LEF lifetime is generally rate-limiting for synapsis (Supplementary Fig. 3). Notably, biological processes that prolong the constraining LEF lifetime will most strongly improve synapsis efficiency since $\tau_{constrained}$ is the denominator for both relative timescales.

Our analytical theory allows us to understand both mechanistically and quantitatively how the various loop extrusion model parameters regulate synapsis efficiency, and reveals their relative impact on synapsis efficiency. While live-cell imaging data[41] helped refine the loop extrusion model described above, additional mechanisms have been suggested by other experimental studies. We, therefore, consider further extensions to our initial loop extrusion model in the next sections.

We systematically investigated the effect of each model extension with our analytical theory and fast 1D simulations, since 3D polymer simulations would be prohibitively slow. The setup for the 1D simulations mirrored 3D polymer simulations (Supplementary Fig. 4a), and we recorded a successful synapsis event (Supplementary Movie 1) when at least one constraining LEF remained and the gap sizes on both sides of a DSB were smaller than ~2kb (a distance where synapsis by passive diffusion is highly efficient[17,55]). Since unconstrained DSBs have synapsis kinetics slower than passive diffusion alone (Fig. 2c, d), a failed synapsis event (Supplementary Movie 2) was logged if all constraining LEFs at a DSB unloaded before the condition for successful synapsis was met (see Supplementary Fig. 4b and Methods). Therefore,

with the methodology above, the synapsis efficiency calculated from 1D simulations estimated the fraction of DSBs with fast synapsis kinetics.

## BE stabilization of LEFs and long-lived LEFs strongly improve synapsis efficiency

Having found that the synapsis efficiency depends most strongly on $\tau_{constrained}$, we first investigated plausible biological mechanisms that would increase the constraining LEF lifetime.

First, we examined to what extent synapsis efficiency could be improved via LEF stabilization by BEs (Fig. 4a, top panel; Supplementary Movie 3; BE stabilization already introduced in Fig. 2b). In mammalian interphase, cohesin and CTCF are the most prominent LEF and BE candidates in vivo, respectively. Recent work has demonstrated that CTCF may stabilize cohesin by protecting cohesin from WAPL-mediated dissociation and/or by facilitating ESCO1-mediated acetylation of cohesin, ranging from fourfold at the *Fbn2* locus to an up to a 20-fold increase in cohesin's residence time in other contexts[41,56,57]. We, therefore, carried out 1D simulations where a LEF bound to a BE exhibits a 1-, 2-, 4-, 8-, or 16-fold increased residence time. By 1D simulations and theory, we found that stabilization of LEFs at BEs strongly increases DSB synapsis efficiency (Fig. 4a and Supplementary Note 1.4.1). The extended theory provides intuition for the process (Supplementary Fig. 5a, b): first, stabilization of LEFs at BEs will improve $P_{constrained}$ through a greater genome coverage by DNA loops; second, stabilization of LEFs at BEs prolongs the window of opportunity for gap-bridging LEFs to load and bridge the gaps. However, stabilization of LEFs at BEs achieves increased synapsis efficiency at the cost of slower synapsis (Supplementary Fig. 5f). Thus, while stabilization of LEFs at BEs strongly improves synapsis efficiency, it was still insufficient (on its own) to achieve the >95% efficiency observed in vivo[26].

Second, we considered having a subpopulation of very long-lived LEFs (Fig. 4b, top panel and Supplementary Movie 4). The most likely LEF candidate, cohesin, exists in multiple forms, and recent work has shown that acetylated cohesin-STAG1 exhibits a much longer residence time than unacetylated cohesin-STAG1[56]. Based on Wutz et al., we estimate that ~30% of all cohesins could be acetylated, and exhibit up to a 50-fold increase in residence time compared with the unacetylated cohesins (Supplementary Note 2.3). Thus, we carried out 1D simulations where a subpopulation of long-lived LEFs (5, 10, or 20%) exhibited a 20-fold increase in processivity, and extended our analytical model accordingly (Supplementary Note 1.4.2). We found—both by 1D simulations and our analytical model—that a small portion of long-lived LEFs improved the synapsis efficiency (Fig. 4b). The close agreement between our theoretical prediction and 1D simulation results supports our mechanistic interpretations of long-lived LEFs' role in synapsis: the long-lived LEFs facilitate synapsis mainly by acting as constraining LEFs (Supplementary Fig. 5d, g) and less frequently as gap-bridging LEFs (Supplementary Fig. 5e, h)[52]; the long-lived constraining LEFs provide a larger time window to attempt synapsis similar to stabilization of LEFs at BEs, as seen in the increased mean-synapsis time (Supplementary Fig. 5i). Yet, once again, while long-lived LEFs strongly improve synapsis outcome, this mechanism is insufficient on its own to achieve the desired (>95%) synapsis efficiency.

To understand the limitations of the two above-proposed mechanisms, we looked at how they separately affect $P_{constrained}$ and $P_{end\text{-}joining|constrained}$. We found that stabilization of LEFs at BEs and long-lived LEFs can realize ~100% $P_{constrained}$ (Supplementary Fig. 5j, k), suggesting that the failure to achieve near-perfect synapsis efficiency is due to inefficient gap-bridging by these mechanisms. We thus turned our attention to additional mechanisms, which could potentially improve $P_{end\text{-}joining|constrained}$, and ultimately increase $P_{synapsis}$.

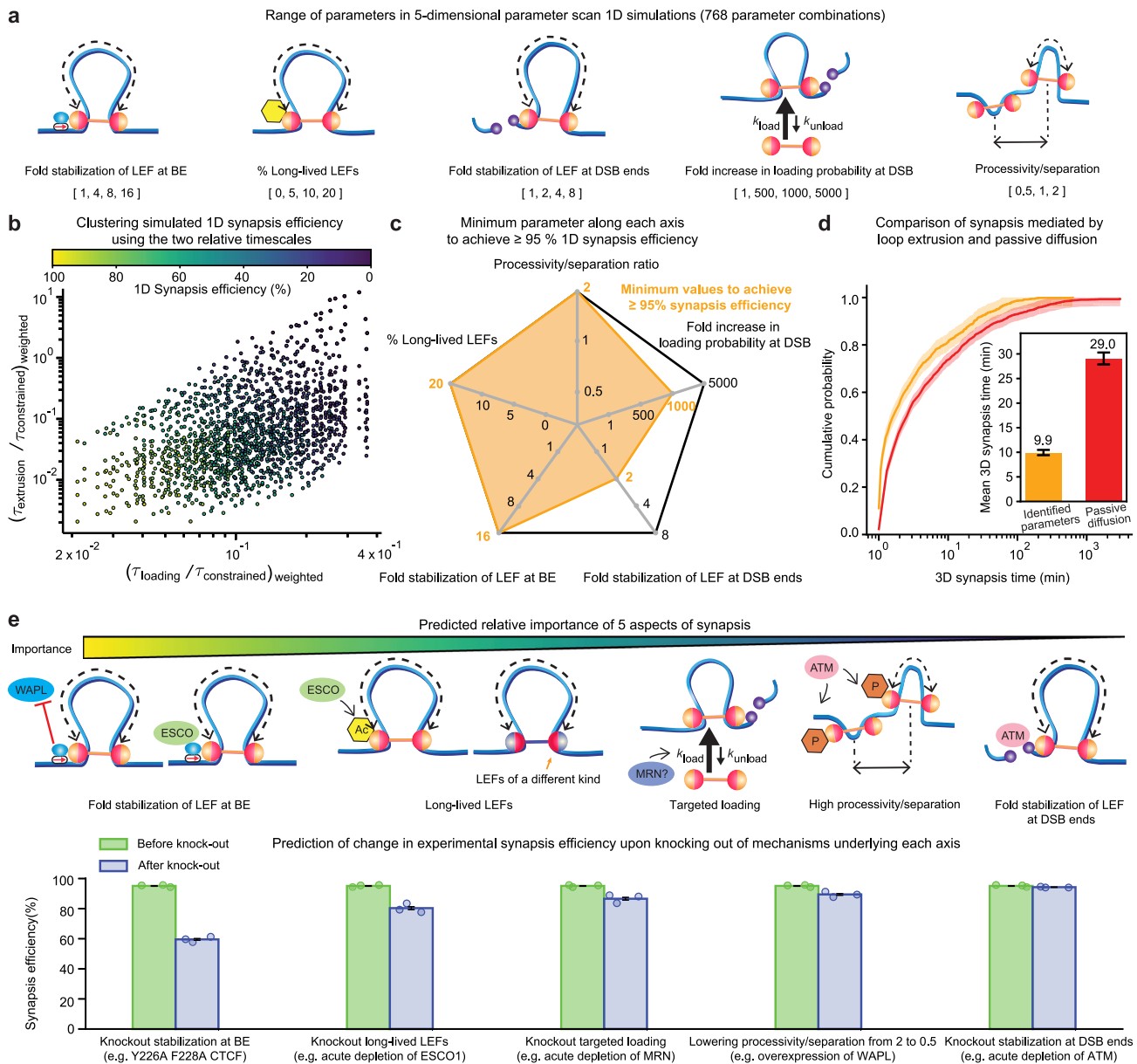

**Fig. 5 | Large-scale 1D simulations reveal a physiologically plausible parameter regime that achieves synapsis with ≥95% efficiency. a** Parameters scanned along each of the five dimensions. **b** The two relative timescales in Fig. 3d can cluster the 5D parameter scan data points based on synapsis efficiency. **c** Among all the parameter combinations that achieve a synapsis efficiency average ≥95% ($n = 3$ independent 1D simulations, with 216–218 DSB events per simulation), we recorded the minimum parameter along each of the five dimensions. **d** Significantly accelerated synapsis with loop extrusion parameters highlighted in (**c**). Cumulative probabilities of synapsis time for synapsis with loop extrusion parameters highlighted in (**c**) and with passive diffusion alone are calculated from 1313 and 2223 DSB events, respectively. The shades around the cumulative probability curves represent the 95% confidence interval of the cumulative probability estimated with

the Dvoretzky–Kiefer–Wolfowitz inequality. The error bars of the bar plot represents a 95% confidence interval of the mean using maximum likelihood estimation of the exponential distribution accounting for censored data[41]. **e** The five aspects of synapsis are ranked ordered based on the predicted reduction in synapsis efficiency upon knocking out the corresponding mechanism. The schematic diagram shows a plausible mechanistic basis for each of the five aspects of synapsis. The bar plot shows the average synapsis efficiency before and after knocking out each of the five mechanisms. The error bars represent the standard error of the mean ($n = 3$ different parameter combinations that achieved ≥95% synapsis efficiency), with the synapsis efficiency from each parameter combination overlaid as individual dots on the bar plot.

## DSB end stabilization of LEFs only modestly improves synapsis efficiency

To identify mechanisms that facilitate the gap-bridging process (i.e., increase $P_{\text{end-joining}|\text{constrained}}$), we examined the necessary conditions that have to be met to achieve synapsis. One key requirement for synapsis is the simultaneous bridging of the gaps on both sides of the DSB. In other words, a gap-bridging LEF that has bridged the gap on one side of the DSB needs to remain on DNA until the gap on the other side of the DSB is also bridged (Supplementary Fig. 5l). Therefore, we

reasoned that factors stabilizing gap-bridging LEFs can help facilitate the synapsis process (Supplementary Movie 5).

Recent experimental work[44] has suggested that cohesins associated with DSB ends are stabilized, resulting in increased processivity (DSB stabilization of LEFs; Fig. 4c, top panel), in line with our hypothesis. We thus extended our theory to account for the DSB stabilization of LEFs (Supplementary Note 1.4.3) and performed 1D simulations where a LEF in contact with a DSB end experiences a 1-, 4-, 8-, 12-, or 16-fold increase in residence time.

We found that while DSB stabilization does lead to improved synapsis outcome, the effect is much milder than the stabilization of LEFs at BEs (Fig. 4c, bottom panel).

We identified two reasons why DSB stabilization of LEFs only modestly improves synapsis efficiency by itself. First, DSB stabilization does not modify $P_{constrained}$ (see Supplementary Note 1.4.3). Second, DSB stabilization only helps stabilize gap-bridging LEFs but does not help the loading of gap-bridging LEFs in the first place.

Together, these results suggest that maintenance of the end products of gap-bridging (i.e., proteins that give gap-bridging LEFs a boost in residence time) is not the bottleneck of synapsis. Instead, this suggests that the establishment of the end products of gap-bridging are the limiting step; this led us to search for mechanisms that accelerate the rate of gap-bridging.

### Targeted loading of LEFs at DSB ends improves synapsis efficiency by accelerating gap-bridging

Guided by the objective of finding mechanisms that accelerate gap-bridging product formation, we turned to biological mechanisms that increase the loading rate of LEFs to the gap. One such mechanism is the targeted loading of LEFs to the DSB ends (Fig. 4d, left panel and Supplementary Movie 6). Experimental support for targeted loading of LEFs at DSB ends comes from recent studies that observed accumulation of cohesin at DSBs sites leading to a ~2–10-fold cohesin enrichment at restriction-enzyme induced DSB sites[44,58,59] possibly mediated by the MRN/MRX (MRN in mammalian cells, MRX in *Saccharomyces cerevisiae*) complex[59], though two other studies reported cohesin enrichment only for S/G2 cells but not G1 cells via laser induction[60,61], suggesting that further experimental work is needed. However, there are also indications that other LEFs may function as gap-bridging LEFs, such as MRE11[62], and evidence of targeted loaded LEFs helping γH2AX spreading near a DSB end[44,62–64].

To test how targeted loading of LEFs to DSB ends aids synapsis, we implemented 1D simulations where loading of LEFs within 1 kb of the DSB is 250-, 500-, 750-, 1000-, 5000-, or 10000-fold more likely than at other similarly-sized genomic loci; this corresponded to about 5, 9, 13, 17, 50, or 67% of LEF loading events occurring at the DSB sites in our simulations. To test the physiological plausibility of these parameter values, we generated ChIP-seq-like data from our 1D simulations for the accumulation of LEFs around the DSBs, and compared this to experimental ChIP-seq data for cohesin accumulation at DSB ends[44]. We found good agreement between the ~2-fold experimental enrichment of cohesin in the DSB-containing TADs and our simulated enrichment of LEFs (with 250X targeted loading) (Supplementary Fig. 6a, b). This demonstrates that even a mild accumulation of cohesin, as seen by a ChIP-seq experiment, may represent a strong loading bias of LEFs at DSB ends.

From our 1D simulations, we found that targeted loading of LEFs significantly improved synapsis efficiency, and that the effect saturated quickly at ~750-fold increased targeted loading (Fig. 4d, middle panel), which corresponds to ~2.5-fold enrichment of LEFs in DSB-containing TADs (Supplementary Fig. 6a). We additionally extended our theoretical model to include the targeted loading mechanism and found consistent results (Fig. 4d, middle panel). Thus, accelerated loading at DSBs makes gap-bridging LEFs more likely to finish synapsis before the constraining LEF unloads (Supplementary Note 1.4.4); this leads to higher synapsis efficiency (Fig. 4d, middle panel) and lower mean-synapsis time compared with the case of no targeted loading (Fig. 4d, right panel). Nevertheless, although targeted loading increases both the speed and efficiency of synapsis and agrees with experimental data, it still falls short of the necessary >95% synapsis efficiency seen in vivo as a mechanism on its own.

### Large-scale 1D simulations identify parameters required for synapsis with near-perfect efficiency

Thus far, we considered four extensions of the simple loop extrusion model, and found that each individually improves synapsis outcomes, but falls short of the >95% synapsis likely required in vivo. We, therefore, asked whether combinations of our proposed mechanisms can achieve the necessary synapsis efficiency. To test this, we carried out a systematic large-scale sweep of all five model parameters, which included (1) the fold stabilization of LEFs at BEs, (2) the fraction of long-lived LEFs, (3) the fold stabilization of LEFs at DSB ends, (4) the fold increase in loading rate at the DSB, (5) the ratio of LEF processivity to LEF separation. This yielded a total of 768 different parameter combinations (Fig. 5a and Supplementary Note 2).

Interestingly, the synapsis efficiency of all 768 parameter combinations was separated neatly along two axes (Fig. 5b) composed of the two relative, weighted timescales $(\tau_{loading}/\tau_{constrained})_{weighted}$ and $(\tau_{extrusion}/\tau_{constrained})_{weighted}$ defined by Eqs. (147)–(148) (Supplementary Note 1.6). This representation of our 1D simulation results demonstrates that despite large mechanistic differences between the models, our theory identifies a universal pair of timescales that captures the central features of the LEF-mediated DSB synapsis process.

We next focused on the models that achieve high synapsis efficiency, and sought to understand more mechanistically how our four proposed extrusion mechanisms may combine for efficient synapsis. Among the parameter combinations that met the 95% efficiency criterion, we plotted the minimum required value of each parameter (Fig. 5c and Supplementary Movie 7). First, we found the minimal required value for fold stabilization of LEFs at BEs was 16-fold, consistent with experimental estimates of up to 20-fold stabilization[56] (see Supplementary Note 2.2). For a fraction of long-lived LEFs, we found 20% by our 1D simulation sweep, and experimentally it is estimated that up to ~30% of cohesins are acetylated (and have a longer DNA-bound residence time; see Supplementary Note 2.3). For the fold stabilization of LEFs at DSB ends, we found the minimal required value was 2, similar to the ~2–4 range suggested by ref. 44 (see Supplementary Note 2.4). For the fold increase in loading probability at DSB, we needed a value of 1000 (resulting in ~17% of LEFs loading at the DSB). Finally, we found that we needed a processivity/separation ratio of 2 (which is within the range suggested by refs. 39,40, see Supplementary Note 2.1). We note that the three models with ≥95% synapsis efficiency achieve synapsis within 12-14 minutes on average (assuming a total extrusion speed of 1 kb/s), consistent with the 6–17 min synapsis time estimated from prior data[20,42,43].

We next performed 3D polymer simulation with the minimal values to achieve ≥95% synapsis efficiency in 1D simulations along each of the five dimensions (highlighted in orange in Fig. 5c). Notably, this parameter combination yielded a mean 3D synapsis time of 9.9 min (Fig. 5d), consistent with the 6–17 min previous estimate. Further, this parameter combination significantly accelerated synapsis (Fig. 5d) so that the 3D synapsis time to reach 95% was reduced to ~40 min (compared with ~160 min for synapsis mediated by passive diffusion alone), with the upper bound of the 95% confidence interval roughly in line with the experimentally determined value of 20 min.

We also generated chromosome conformation capture (Hi-C)-like contact maps from 1D simulations with the models, which achieve high synapsis efficiency. Consistent with the experimental Hi-C maps reported by ref. 44, we find that DSBs result in an X-shaped stripe pattern in the vicinity of the break site, whose size of ~1 Mb is also similar to the experimental counterpart (Supplementary Fig. 7a). Moreover, these results suggest that in order to capture the temporal dynamic changes to 3D genome structure caused by DSB formation, it is necessary to perform Hi-C at shorter time-intervals post-DSB formation (e.g., at 5 min intervals), and points to the need to have fast mechanisms of inducing DSBs at specified genomic locations to better study the effect of DSB end synapsis on 3D genome organization[62].

In summary, our simulations show that with experimentally plausible parameter values, loop extrusion can achieve fast and ≥95% efficient synapsis. We thus propose that loop extrusion plays a previously unrecognized role in mediating DSB synapsis as part of NHEJ in mammalian cells.

## Discussion

Synapsis is the first step of DSB repair by NHEJ, which is the dominant repair pathway in the G1-phase. Synapsis has largely been assumed to occur by passive 3D diffusion[65]. However, our 3D polymer simulations show that passive diffusion would lead to unphysiologically slow synapsis in mammalian nuclei.

Here we propose that protein-mediated DNA loop extrusion may promote fast and efficient synapsis in cells. We emphasize that loop extrusion can be a preemptive mechanism facilitating synapsis, in contrast to the reactive recruitment of DSB repair machinery after the DSB has occurred, where DSB ends may diffuse apart during the recruitment period. We built a probabilistic theoretical framework to understand LEF-mediated DSB end synapsis. The loop extrusion model with dynamic parameters confined by experimental data measured at the *Fbn2* locus fell short of the synapsis speed observed in vivo, but did accelerate synapsis compared with passive diffusion alone. Guided by our analytical theory, we explored four plausible extensions to the loop extrusion model that constituted mechanistically distinct ways to improve synapsis outcome and tested the theory with 1D simulations of the extrusion-mediated DSB synapsis process, finding they were in good agreement. We found that while each mechanistic extension could moderately improve synapsis efficiency, it was by combining all four mechanisms that we found a regime with synapsis kinetics consistent with experimental observations. Our theory demonstrates that loop extrusion is a viable and efficient way to mediate the first step of the NHEJ process, DSB end synapsis, by co-opting the cell's chromosome organization machinery. Altogether, the action of constraining LEFs and gap-bridging LEFs may constitute a new paradigm for thinking about the process of DSB end synapsis.

A broader role for loop extrusion in DNA repair is beginning to emerge. Recent experimental studies have proposed that DNA loop extrusion may facilitate DSB repair foci formation by mediating γH2AX spreading[44,62–64] and by forming structural scaffolds with 53BP1 and RIF1 to protect DSB ends from aberrant processing[66]. In addition, loop extrusion by cohesins (one of the best-studied LEFs) has been proposed to facilitate V(D)J recombination[67–69] and class switch recombination (CSR)[70] by aligning the genomic loci to be recombined. It is worth noting that loop extrusion's role in V(D)J recombination likely precedes DSB occurrence[67–69], since the DSB ends are already in proximity when DSB occurs and are kept close by the post-cleavage complex along and other DDR repair factors[71]. Our model can be readily generalized to encompass CSR, since CSR can be considered a special case of NHEJ synapsis where two instead of one DSBs are induced[11,70]. In addition, the proposed role of LEFs in mediating γH2AX spreading[44,62–64] is synergistic with our proposed gab-bridging mechanism. Our work, therefore, provides a framework to extend, integrate, and test several models.

Importantly, we note some limitations of our study. First, we have not exhausted the space of possible mechanisms that could facilitate DSB end synapsis. For example, we do not consider alternative extensions of loop extrusion based on the diffusive sliding of LEFs[72]. Furthermore, it would be interesting to incorporate LEF bypass, as recent work has shown that condensins can bypass each other[73,74], which would help improve genome coverage by DNA loops. We have also not considered the effect of DNA-damage response (DDR) foci[44,75–80] on synapsis. Repair factors recruited to the DDR foci, such as 53BP1, could drive microphase separation to help constrain the DSB ends and promote synapsis[81–83]. The shift to a repressive chromatin state at the DSB through the accumulation of repressive complexes,

including HP1, might also help constrain the two DSB ends to facilitate synapsis[84]. Other chromatin-binding proteins, such as multivalent transcription factors, could also constrain DSB end movement by bridging two sides of a DSB[85]. The long protein filaments formed around DSB ends by downstream NHEJ factors XRCC4, XLF, and DNA ligase IV[86–89] could also facilitate dynamic synapsis[20,90], potentially by reducing the dimensionality of the system and allowing bridging of DSB ends at a larger distance[20,88,89]. Such dynamic synapsis structures at DSB sites could efficiently transition to NHEJ ligation products[20,91,92]. Enhanced mobility of chromatin around DSBs mediated by nuclear actin and microtubules has also been hypothesized to promote synapsis[93,94]. We have assumed an extrusion speed of 1 kb/s based on in vitro studies[28–30,36,95], though we note that the in vivo extrusion speed is associated with large uncertainty and was estimated to be ~0.2–0.25 kb/s for cohesin at the *Fbn2* locus in mESCs[41]. Furthermore, the extrusion speed may vary with different genomic loci, cell types, and LEF types, and may change upon the recruitment of DSB repair factors. Finally, we have assumed a uniform distribution of LEFs throughout the genome, while previous work suggests that cohesin loading may not be uniform across the genome[35,96–99]. In summary, a limitation of our study is that it does not account for all possible mechanisms. Instead, our focus here is to specifically evaluate if and how loop extrusion can mediate synapsis and contribute above and beyond passive 3D diffusion. Nevertheless, as our detailed quantitative knowledge of these additional mechanisms improves, it will be important to extend our model to incorporate these additional mechanisms in the future.

Here, we speculate on candidate biological processes that could execute the required tasks for (1) stabilization of LEFs at BEs, (2) creating a mixture of long-lived and short-lived LEFs, (3) generating high LEF processivity/separation ratio, (4) facilitating targeted loading, and (5) DSB stabilization. First, we suggest that the stabilization of LEFs at BEs could be mediated by CTCF-mediated cohesin protection from its unloader WAPL[57] or ESCO-mediated acetylation[56]. Second, we suggest that the LEF processivity/separation ratio may increase following a DSB, perhaps through ATM-mediated phosphorylation of cohesin subunits SMC1 and SMC3[100], consistent with the global stabilization of cohesins observed experimentally[44]. Third, long-lived LEFs may correspond to acetylated STAG1-cohesins[56], or LEFs of a different kind that exhibits higher processivity. Fourth, we suggest that targeted LEF loading at DSBs may be mediated by the MRN/MRX complex as the knockdown of MRX led to a significant decrease in cohesin loaded at DSBs[59]. Importantly, given MRN's hypothesized loop extrusion activity[101] and the enrichment of MRN subunit NBS1 at DSB sites[102], MRN itself could also function as gap-bridging LEFs, effectively achieving targeting loading of LEFs while alleviating the required fold increase in loading of cohesin at DSBs. Fifth, we speculate that LEF stabilization by DSBs may be mediated by the ATM complex, which phosphorylates cohesins and accumulates cohesins in the DSB-containing TAD[44,103]. Finally, we note that our mechanistic understanding of loop extrusion and DSB repair is advancing rapidly, such that other factors and mechanisms are likely to be found to play a role beyond the ones mentioned above.

To facilitate the experimental testing of our model, we used our theory and 1D simulations to make specific and quantitative predictions (assuming that cohesin and CTCF play the main role of LEF and BE; Fig. 5e and Supplementary Fig. 7b). Our theory predicts that reducing $\tau_{constrained}$ would most strongly decrease synapsis efficiency. Indeed, our 1D simulations predict the loss of stabilization of LEFs at BEs to have the strongest effect: loss of stabilization of LEFs at BEs would reduce LEF-mediated synapsis efficiency to ~60%. Experimentally, this may be tested by mutating Y226 and F228 in the N-terminus of CTCF since this is predicted to eliminate CTCF-mediated stabilization of cohesin without affecting CTCF binding to DNA[57]. Next, we predict that eliminating long-lived LEFs would have the

second-strongest effect, reducing LEF-mediated synapsis efficiency to ~80%. For example, this could be achieved through acute auxin-inducible degron (AID) depletion of ESCO1 or STAG1. Knocking out targeted loading would have the third-strongest effect, reducing LEF-mediated synapsis efficiency to ~87% and drastically increasing the mean-synapsis time by ~181%. Experimentally, acute depletion of MRN would be one way of testing this. Lastly, although we predict lowering the processivity/separation ratio from 2 to 0.5 and knock-out of DSB stabilization to be relatively mild, reducing LEF-mediated synapsis efficiency to ~89 and ~94%, respectively, their knock-out would slow down synapsis substantially increasing the mean-synapsis time by ~67 and ~39%, respectively. Finally, given the high redundancy between the mechanisms considered in our study, we also predicted the quantitative effect of double knock-outs and alteration of the extrusion processivity/separation (Supplementary Fig. 7b). Consistent with our model, experimental depletion of cohesin subunit RAD21 led to significantly increased chromosomal translocations[104]. Note that our 1D simulations and analytical theory do not account for passive diffusion and other mechanisms acting in parallel to loop extrusion, and thus our prediction should be taken as an upper bound for the change in synapsis kinetics and efficiency.

Broken DNA-end synapsis is a key but understudied step in DSB repair. Our theory provides a new framework for rethinking this initial step of the NHEJ process in the context of our current understanding of 3D chromosome organization by loop extrusion. In summary, we predict that DNA loop extrusion plays a previously underappreciated role in DNA repair by mediating DNA double-strand break synapsis.

## Methods

### Time steps and lattice set-up
We used a fixed-time-step Monte Carlo algorithm for 1D simulations as described in previous work[53]. Each lattice site corresponded to 1 kb of DNA. For 3D polymer simulations, we defined a single chromosome consisting of 70,000 lattice sites (corresponding to 70 Mbp of DNA); for 1D simulations, we defined a single long chromosome as a lattice of $G = 2,164,800$ sites (corresponding to 2164.8 Mbp of DNA). Loop extruding factors (LEFs) were comprised of two motor subunits that move bidirectionally away from each other one lattice site at a time. Like most cohesin simulations[37], we assumed LEFs cannot bypass each other upon encounter. LEFs also could not extrude past the first and the last lattice sites, so LEFs could not "walk off" the chromosome.

### Boundary elements
Each boundary element (BE) occupied one lattice site. BEs were directional (indicated by the red arrow in schematics) and only if the LEF motor subunit's extrusion direction was convergent with the direction of a BE, would the motor subunit be stalled by the BE with a probability equivalent to boundary strength $b$. Unless specified otherwise, a boundary strength of $b = 0.5$ was used, in line with experimental estimates of CTCF binding site occupancy[39,40]. Once a motor subunit was stalled by a BE (i.e., the subunit was stopped at the BE lattice site), no further movement of the subunit was allowed until the LEF dissociated from the locus and reloaded back onto the chromosome somewhere else. Only one motor subunit could occupy a BE lattice site at a time. We placed pairs of BEs on the chromosome (each pair consisting of two divergently oriented BEs) with the spacing between each pair ranging from 200 to 1200 kb Fig. 2a, since TAD sizes range from 200 kb to 2.5 Mb based on experimental data[105,106].

### DSB sites
Each DSB site occupied two lattice sites on the chromosome, each of which corresponded to a DSB end. We introduced DSBs approximately every 3 Mb for 3D polymer simulations and 10 Mb for 1D simulations. We first randomly picked the DSB site in the very first TAD on the chromosome, and then we found the TAD 3/10 Mb to the right of the

first DSB site, and randomly induced the second DSB in the TAD (so that the distance between DSBs and BEs were randomized), and so forth. This results in altogether 22–28 DSB sites for 3D polymer simulations and 216–218 DSB sites for 1D simulations on the chromosomes. We first ran 100 thousand time steps of the 1D simulations, and then introduced all the DSBs simultaneously. After DSB occurrence, LEFs were not allowed to extrude past DSB ends. We allowed multiple motor subunits to occupy the DSB end lattice site at the same time, to enable simulations with targeted loading of LEFs to DSB sites.

### LEF association and dissociation rates
All 1D simulations were performed with a fixed number of LEFs, determined by the ratio of chromosome length $G$ and LEF separation $d$ (i.e., the inverse of LEF density). The dissociation rate was linked to the LEF processivity $\lambda$ (i.e., the average length of DNA extruded by an unobstructed LEF before it dissociates) (Supplementary Note 1.3.2). After a LEF dissociated from the chromosome, it immediately and randomly reloaded onto a lattice position on the chromosome that was not occupied by other LEFs' motor subunits. The only lattice sites where co-occupancy and loading of multiple LEFs were allowed were at DSB ends.

### 3D polymer simulations via OpenMM
We performed 3D polymer simulations using Polychrom[107], which wraps the molecular dynamics simulation toolkit OpenMM[108]. LEFs were simulated as harmonic bonds between two chromosomal monomers. To simulate the dynamics of LEFs, we first performed 1D simulations, and ported the loop extrusion dynamics from the 1D simulations to the 3D simulations. The underlying 1D simulations were first run for 10,000 translocation steps to reach a steady state before being coupled with 3D polymer simulations.

The chromosome used in 3D polymer simulations consisted of 70,000 consecutive monomers bonded with pairwise potential

$$U_{\text{bond}}(r) = \frac{k}{2}(r - b_o)^2 \tag{1}$$

where $k = 2k_bT/\delta^2$ is the spring constant ($k_b$: Boltzmann constant, T: temperature, $\delta$: 0.1 monomer), $r$ is the 3D distance between adjacent monomers, and $b_o$ is one monomer size (the mean 3D distance between adjacent monomer). Monomers connected by LEFs were bonded via the same potential. The excluded volume interactions between monomers were implemented with a weak polynomial repulsive potential

$$U_{\text{exc}}(r) = \frac{\epsilon_{\text{exc}}}{\epsilon_m}\left(\frac{r\sqrt{6/7}}{\sigma}\right)^{12}\left(\left(\frac{r\sqrt{6/7}}{\sigma}\right)^2 - 1\right) + \epsilon_{\text{exc}} \tag{2}$$

for $r < \sigma = 1.05$ monomer, where $\epsilon_m = 46,656/823,543$ and $\epsilon_{\text{exc}} = 50k_bT$.

For each simulation run, the polymer was initialized as a compact conformation on a cubic lattice with normally distributed velocities. The error tolerance for the variable time step Langevin integrator was set to 0.01, and the collision rate was set to 1. Simulations were performed with spherical confinement to achieve a DNA volume fraction of 20% per simulation volume. The initial cubic lattice conformation was allowed to relax into a steady-state conformation for 15 million simulation time steps. Subsequently, harmonic bonds between LEFs were added (where LEF starting positions were taken from a steady-state loop extrusion 1D simulation), and we performed a further local energy minimization to relax the structure with the LEFs constraints. We allowed for at least 173,690 additional simulation steps for the new conformations to relax with the LEF constraints before introducing DSBs (i.e. removing the harmonic bond between the two monomers corresponding to the DSB ends).

The simulation time and distances were calibrated in real-time and distances using MSDs determined from microscopy data[41]. Using inference methods described previously in ref. 41, we inferred that each simulation time step corresponds to ~0.014 s and each monomer corresponds to ~30 nm in diameter. We used a total LEF extrusion speed of 1 kb/s on chromatin (0.5 kb/s for each motor subunit)[28–30,36,95]. The chromatin diffusion coefficient was $0.7 \times 10^{-3}$ μm²/s[41]. We note there is a wide range of estimates for the chromatin diffusion coefficient[109–111], and the value used in this study is consistent with estimates on the order of $1 \times 10^{-3}$ μm²/s based on recent live-cell imaging experiments in mammalian cells[41,112].

### Monitoring of synapsis events
For the 3D polymer simulations, we waited for the equivalent of 1 min of real-time (4286 simulation time steps) to allow the DSB ends to diffuse freely, and then started to record synapsis events. We motivated the 1-min time scale as the mean expected time for the recruitment of DSB repair factors such as Ku70/80, which are necessary to facilitate synapsis[16,42,65,113]. We then calculated the 3D distance between the two DSB ends at every simulation time step following the initial 1-min lag and recorded a synapsis event when the two DSB ends were within our contact radius. Given the average DSB end displacement per time step was around 1 to 2 monomers, we used a contact radius of four monomers unless otherwise specified so that we did not miss any synapsis events. Similar trends were observed with capture radii of three or five monomers (Supplementary Fig. 8).

For the 1D simulations that were not coupled to 3D polymer simulations, at every time step during the simulation (and for each DSB site), we first checked whether there was at least one LEF whose two motor subunits were on opposite sides of the DSB (i.e., whether the DSB site was constrained by at least one LEF). For the constrained DSB sites, we counted the number of (unextruded) lattice sites between the innermost constraining LEF and the DSB ends (i.e., the gap size); if there were only two lattice sites (corresponding to 2 kb DNA), we considered synapsis was achieved and we stopped monitoring this site. We scored all DSB sites that were not initially constrained by LEFs as having failed to achieve synapsis, and did not continue monitoring these sites in later time steps (Supplementary Fig. 4b).

Due to the coarse-graining necessary for efficient simulations, we used a synapsis threshold of ~2 kb, which is larger than the actual distance between the two DSB ends held together by the synaptic complex[114]. However, when the length of unextruded DNA is on the order of 2 kb, the two DSB ends diffusion is so efficient that the DSB ends will be aligned within seconds for downstream ligation[17,55], thus making a negligible contribution to the synapsis time.

### Modifications to LEF dynamics with additional mechanisms
With the stabilization of LEFs at BEs, the LEFs with at least one motor subunit at the lattice sites representing BE had $w$-fold reduction in dissociation rate. With the stabilization of LEF at DSB ends, the LEFs with at least one motor subunit at the lattice sites representing DSB ends had $r$-fold reduction in dissociation rate. With a small fraction $\alpha_o$ of long-lived LEFs, the long-lived LEFs had a dissociation rate 20-fold smaller than the normal LEFs (non-long-lived LEFs). When there was a subpopulation of long-lived LEFs, the separation $d$ referred to the separation of long-lived LEFs and normal LEFs combined. With targeted loading of LEFs at DSB, the loading probability at the lattice sites representing DSB ends was $F$-fold higher than anywhere else on the chromosome.

### Simulated ChIP-seq data
We first divided the genome into 5-kb bins, and then we counted the number of LEF motor subunits in each bin using the stored LEF positions 10-min post-DSB, and wrote this to a BED file containing each bin's score (i.e., the number of LEFs). Along with another BED file containing the DSB coordinates, we used the plotHeatmap command from deepTools[115] to generate the ChIP-seq heatmaps shown in Supplementary Fig. 6b. For the fold enrichment calculated in Supplementary Fig. 6a, we counted the number of LEF motor subunits in each DSB-containing TAD across the different time points, and finally normalized the average LEF subunit counts post-DSB by the corresponding average LEF subunit counts before DSB occurrence. The boundaries of the chr20 DSB-containing TAD of DIva cells were determined by CTCF binding sites adjacent to the DSB site using CTCF ChIP-seq data[44].

### Simulated Hi-C contact maps
We first normalized the LEF positions on the lattice sites by subtracting the positions of the closest DSB sites, so that the LEF positions were all relative to the closest DSB sites. We only included LEFs that were within ±2.5 Mb of the DSB sites. We then calculated the contact probability maps directly from the LEF positions, by utilizing a Gaussian approximation developed previously to simulate bacterial Hi-C maps[53]. An iterative correction was then applied to the calculated contact maps to generate the final contact maps[116]. Note in our 1D simulations, unlike experiments, targeted loading of LEFs to DSB ends continued even if they are already synapsed (this was so that the LEF abundance at each DSB site did not depend on when the DSB sites are synapsed). Moreover, in experiments, the stripe pattern measured by Hi-C[44] might be weaker because certain DSBs sites could have already been synapsed/repaired in a fraction of the cells, whereas we did not simulate the DNA repair process downstream of synapsis.

### Reporting summary
Further information on research design is available in the Nature Portfolio Reporting Summary linked to this article.

## Data availability
Simulation data were available in the GitHub repository https://github.com/ahansenlab/DNA_break_synapsis_models/tree/main/Data. Analytical theory formulation is available at https://github.com/ahansenlab/DNA_break_synapsis_models/tree/master/Mathematica.

## Code availability
Simulation and analysis codes, as well as analytical theory formulated in Mathematica are available in the GitHub repository https://doi.org/10.5281/zenodo.7677969[117].

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

## Acknowledgements
We thank Dr. Leonid Mirny, Dr. Ed Banigan, Dr. Joe Loparo, Dr. Thomas Graham, Dr. Assaf Amitai, Dr. Kyoko Yokomori, Dr. Miles Huseyin, Sarah Nemsick, Henrik Pinholt, and the Hansen lab for insightful discussions and comments on the manuscript. J.H.Y. was supported by a MathWorks Engineering Fellowship and a graduate fellowship from the Ludwig Center at MIT's Koch Institute for Integrative Cancer Research. This work was supported by the National Institutes of Health (grant numbers R00GM130896 and DP2GM140938).

## Author contributions
H.B.B. conceived of the project; J.H.Y., H.B.B., and A.S.H. designed the project; J.H.Y. performed the research with joint guidance from H.B.B. and A.S.H.; J.H.Y. and H.B.B. developed the theory and simulation codes with input from A.S.H.; J.H.Y. performed the simulations and analyses with input from H.B.B. and A.S.H.; J.H.Y. drafted the manuscript with input from H.B.B. and A.S.H. and all authors edited the manuscript.

## Competing interests
The authors declare no competing interests.
