## [Peer Review File · Nature Communications]

DNA double-strand break end synapsis by DNA loop extrusionREVIEWER COMMENTS

Reviewer #1 (Remarks to the Author):

This is a potentially interesting contribution on the effect of loop extrusion on synapsis formation following a double-stranded break. The idea can stimulate discussion within the biophysics community interested in chromatin and DNA organisation and function. However I cannot recommend the current version of the manuscript for publication, for the following reasons.

1. The simulations are done with a purely 1D model for loop extrusion, and synapsis formation is essentially a first-passage time problem. While interesting I think this is a significant limitation, and it would be much more convincing to study at least some cases with 3D polymer simulation models where the polymer dynamics is considered (at least to prove when polymer dynamics is not relevant). This is important, I feel, because there are many biophysical details which are included in the modelling (targeted loading, different cohesin populations, LEF stabilisation) and which is not clear to me should be any more important than 3D effects such as polymer end diffusion, polymer concentration and stickiness for instance of DSB for each other or for the rest of the polymer. I understand it is not practical or required for this first contribution to consider 3D effects systematically, but at least the basic case should be done with full 3D polymer dynamics to make the study more convincing.

2. The presentation of the results is difficult to follow, partially because all results, both positive and negative, are discussed in turn. I think the discussion of the subsequent mechanistic model improvement/modifications should be ideally more concise: a brief discussion of the biophysical reason for failure/partial success is perhaps OK, but too much technical discussion here is not so helpful in my view. The analytical theory is nice, but again can be presented more concisely I feel. The current presentation is more suitable for a more specialised journal (Biophys. J. for instance). The videos are I think more clear. They are very useful however they give me an overall impression that there are quite a few parameters which need to be tuned to get the desired effect.

As minor comments:

1. The discussion of criteria for synapsis failure/formation given in the method section (page 18, Monitoring of synapsis events) is not too clear. Given their key relevance to the results, I think these criteria should be briefly clarified in the main text and a diagram here would help (along the lines of the modelling videos which are clearer).

2. I think it would be good to clarify from the beginning that the main model is purely 1D.

3. Did the authors consider what would happen for diffusing LEFs (e.g., Brackley et al., PRL 119, 138101 (2017))? This may be a useful/interesting case to consider to show whether active extrusion is crucial. Targeted loading of diffusing LEFs also leads to their ratcheting which may help diffusing LE as a potential mechanism.

4. In reality there are foci formation at sites of repair which I believe include sites of DSBs. There are chromatin-binding proteins which bridge the DNA and cause microphase separation of heterochromatin and euchromatin and they are likely to have a strong effect on foci formation. Likewise, I think the effect of bridging proteins (e.g., HP1 or other multivalent transcription factors) on synapsis formation is potentially important and should be discussed as an alternative mechanism to look for, beyond what is considered by current model. I feel this is particularly relevant for active or silenced chromatin, less so for inert chromatin, for which this LE-mediated mechanism may be more relevant.

5. While there is a nice discussion of the potential effect of knockout/degrons on future experiments, it

would be good to provide some more quantitative comparison with existing experimental data on DSB. For instance ChIP-seq and Hi-C are discussed in the Supplementary Information but it would be good to have a metric for agreement with experiments and an idea of how much each experiment constrains parameter values in the model.

Reviewer #2 (Remarks to the Author):

Yang, Brandao and Hansen constructed a mathematical model to test the intuitive hypothesis that loop extrusion may aid in synopsis of double-strand breaks. The paper is well written and guides the reader through the thought-process step by step. The figures and movies are clear and very helpful in explaining the research. In short, the form factor and style are excellent, and the authors provide a detailed explanation of their model in the Supplemental Information for transparency.

My main assessment of this manuscript relates to the assumptions the authors have used as the basis for their model, and the relevance of this study to previous work on DSB repair and synopsis. Overall, the study presents interesting elements that might provide some modeling framework for DSB repair, but I find that many of the underlining assumptions and literatures that the authors used to build their model are limited, which results in a confirmatory model that provides a reductive view of DSB repair. Because the current work is lacking critical elements that are required for relating any DSB repair model to established experimental work I find that the study in its current state is not appropriate for publication. The main points are described below, and would require the authors to preform complete revisions of their model and manuscript.

Major points:

1. The authors sought to address the synopsis step of DSB repair via NHEJ (and also alternative NHEJ), but have omitted a large body of literature, while at the same time their studies rely only on references that seems to support their specific narrative and model. This sort of selectivity undermines the validity of their key assumptions, conclusions and thereby the entire premise of this study. In the introduction, and throughout the manuscript, the authors state that it is generally assumed that synopsis occurs via passive 3D diffusion, referencing a review paper on NHEJ repair (Zao et al). This review provides a comprehensive survey on the molecular drivers of NHEJ repair, but contains little to no information on models for diffusion and synopsis.

The authors then claim that 3D diffusion is not efficient enough to promote DSB end synopsis in mammalian cells, to support this they use their model with parameter derived from modeling data of a single study (Taleei et al, 2012), but provide no justification of basing the entire model on a single study, while neglecting to mention many relevant studies where repair dynamics have been measured and modeled (for example, Reynold et al NAR 2012, Yue Liu et al JCS 2019). There are several other studies where the rate of repair by NHEJ of different DSBs have been measured or estimated by various methods. There are also multiple biochemical studies that examined the rate of repair products and derived the kinetics of synopsis. None of these results are being considered in the model, when the authors conclude that diffusion cannot facilitate efficient synopsis.

2. The authors' main motivation for proposing and modeling that synopsis is driven by loop-extrusion is their conclusion that diffusion alone cannot be accounted for the known rates of productive synopsis and repair. They made this conclusion by calculating the encounter of the two sides of the DSB under 3D diffusion, while assuming that the reactive elements that promote synopsis are only localized at broken DNA ends. This assumption is very reductive and contrast many studies that demonstrated that the DNA ends serve as an initiating point for signaling and the recruitment of extended protein networks and formation of a repair foci. DSB induction leads to rapid changes to chromatin around the break and recruitment of a large protein complexes, such as 53BP1, driving phase separation to form condensates thereby extending the regions of reactivity well beyond the broken DNA ends. It is likely that 53BP1 can drive synopsis of the two ends enclosed within the foci, as it was shown to promote

synapsis of neighboring and distal DSBs (Difilippantonio et al Nature 2008, Dimitrova et al Nature 2008).

3. Beyond the properties of 53BP1 and the repair foci, the authors fail to include or even consider other key proteins of the NHEJ pathways that have no enzymatic role, and were shown to serve as scaffolding proteins that stimulate repair by enhancing synapsis. In particular, the authors would benefit from familiarizing themselves with many relevant studies showing that XLF, XRCC4 and LIG4 form extended structures (Hammel et al, Structures 2010, Ropars et al PNAS 2011, Andres et al NAR 2012, Brandi et al Biochem Cell Biol 2013) and that these structures can support dynamic synapsis away from the ends (Reid et al PNAS 2015, Brouwer et al Nature 2016). Other studies also showed that such synapsis can later efficiently transition to ligation products (Reid et al NAR 2017, Conlin et al Cell Reports 2017, Nemoz et al NSMB 2018). Importantly, it was demonstrated that these synapsis structures form at DSBs sites in cells. Overall, several studies, including some that are referenced above have proposed that the elongated structures that form in synapsis can enhance the rate of repair by reducing the dimensionality of the system and diffusion of the ends (Andres et al NAR 2012, Brandi et al Biochem Cell Biol 2013, Reid et al PNAS 2015). This concept was recently described for bacterial DNA repair (Wiktor et al Nature 2021) and likely presents an appropriate landscape for modeling many of the kinetics that occur in biological systems.

4. The authors only consider a scenario whereby a synaptic complex form with the DNA-PK holoenzyme, but such a complex likely occurs as a final step prior to transition to the ligation step. In particular, it was shown that DNA-PKcs is dispensable for synapsis (Brouwer et al Nature 2016, Zhao et al Nature Comm 2019) and its specific roles during cellular repair might occur well after synapsis (Reynold et al NAR 2012, Yue Liu et al JCS 2019).

5. The authors should consider several additional active mechanisms that might contribute to DSB mobility, synapsis and repair, such as the involvement of nuclear actin and microtubules that were shown to affect repair (Lottersberger et al Cell 2015, Schrank BR Nature 2018, Zhu et al Elife 2020).

6. In modeling of loop extrusion rates, the authors consider several studies where the rates were measured in-vitro using naked DNA. These rates would be considerably different for chromosomal DNA in cells, and will likely be affected by the location of the DSB, and by the formation of DSB foci and recruitment of other chromatin factors. The authors should consider how to address these factors in their models.

Additional points:

1. The author studied the LEF separation from 62.5 kb to 500 kb and found better synapsis efficiency with the lowest separation. Later, a separation of 125 kb was used for further simulation. The reason why 125 kb was chosen for further simulation is not apparent.

2. The authors need to provide a physical basis as to why LEFs may be concentrated near the DSB sites in G1 phase.

3. In figure 4B, does the yellow hexagon represent acetylation? It should be noted in the figure or the legend.

4. Figure 4C / line 273: the authors claim that DSB stabilization leads to mildly improved synapsis. Given the position of the data points in Figure 4C, this interpretation seems to be a stretch. The data points basically form a flat line.

5. The authors point out recent studies that show LEFs can bypass each other. This undercuts one of the basic assumptions made in this paper.

6. Line 183: I am not sure if 28 is the correct reference.

7. When I downloaded the zipped software from Github, I was unable to extract it completely due to paths that were too long. The problem seems to be some files in the Data folder. I hope the authors can fix this.

Reviewer #3 (Remarks to the Author):

In this manuscript, Yang and coworkers present a theoretical study of DNA double-strand break (DSB) end synapsis and how loop extrusion would influence the synapsis efficiency and synapsis time.

The authors argue that passive thermal diffusion is not sufficient to bring back the two ends together to rejoin when a double-strand break occurs. They suggest that loop extrusion could be a way for the ends to come together within the biological timescales. They first examine a simple loop extrusion model and find that the efficiency is unsatisfactory. Then they add certain additional features into the model and argue that a combination of these factors would make the synapsis via extrusion highly efficient. They do simulation and also provide analytical arguments to support their claim.

This is indeed an interesting work. The theory and simulations are indeed valid. However, there are some concerns that need to be addressed to make it realistic and publishable in Nature Communications. The concerns are:

1) The authors assume that loop extrusion or loop associated proteins (boundary elements like CTCF, LEFs/cohesin) are the only players in this process. However, we know that there are many other proteins and interactions that hold the chromatin together.

1a) We know that inter-nucleosome interactions do happen, and they can hold chromatin regions together, preventing them from diffusing away.

1b) We know that proteins like HP1, Polycomb complexes, etc., hold chromatin together.

1c) Even in the active regions, proteins like FOXA1 etc bind between different chromatin segments, effectively preventing them from diffusing away.

1d) Equivalently, there is also the micro "phase separation" phenomenon in parallel with loop extrusion; wouldn't that also be important in deciding the efficiency?

Why would an estimate without accounting for any of the factors above be accurate?

2) The authors say that their one bead is 1kb of chromatin. The relevant length scale for DSB and its resolution is 1bp-10bp. In 1kb, there are about 5 nucleosomes, and it is a large region (compared to the bp length-scale relevant for DSB). As mentioned in (1a) above, inter-nucleosomal interaction, etc, would already play a role within that 1kb "bead". So why would a model without basepair resolution be relevant for studying DSB?

2a) The authors take 2million+ lattice sites with 1kb length suggesting a total length of 2billion+ bp. What is the relevance? The longest human chromosome is of the order of 250 Mbps, right?. How are these numbers relevant?

3) It is a limitation that the events are happening in 3D, and the model in this manuscript is a 1D model. It is acceptable as a physics model for loop extrusion estimates. Would the answer be comparable if a similar work is done with a 3D model?

4) The 3D diffusion is taken to be of the order of 10^{-3} micron² per second based on experiments.

4a) One would naively presume that the experimentally obtained diffusion coefficient value will already have the effect of loop extrusion, LEFs etc in it. Why would one need to add these separately?

4b) There is a wide range of estimates for diffusion coefficient. For example, the Maeshima and other labs' experiments seem to suggest that the motion is subdiffusive, and the mean square displacement is about 10^{-2} micron² in 0.1 seconds (this would be like 0.1 micron² per second) [eg. Nozaki et

al. and other other papers]

5) It is found that the boundary strength has little effect. It is not clear how BEs are placed (with orientation etc?). It is written that "We place BEs on the chromosome so that TADs of the sizes shown in Fig. 2A are achieved". However, it is not clear exactly how this is done. Is there an algorithm to place BES to achieve a certain TAD size?

6) The underlying assumption is that all regions of the heterogeneous chromosome will have a similar distribution of LEFs/extrusion etc. However, things could be very different in heterochromatin and euchromatin regions. Is this result valid for both heterochromatin and euchromatin? DSB could happen anywhere, and the timescale could be very different in heterochromatin and euchromatin regions.

6) Why the iterative correction is applied to the simulated data? Typically iterative correction is to eliminate various biases and limitations in experiments, right? Everything, including normalization, is clear in a simulation, and there is no unknown information as far as I understand.

Response to Reviews

We thank the reviewers for their valuable comments, insights, and questions. We have carefully reviewed and addressed all reviewers' comments. The revised manuscript incorporates multiple new 3D polymer simulations and discussions of our model in the broader context of prior literature and alternative mechanisms. We have also improved the presentation of the context and results as per reviewers' comments. Overall, we believe that this revised manuscript is greatly strengthened and fully addresses the reviewers' comments.

Response to Reviewer 1

This is a potentially interesting contribution on the effect of loop extrusion on synapsis formation following a double-stranded break. The idea can stimulate discussion within the biophysics community interested in chromatin and DNA organization and function. However I cannot recommend the current version of the manuscript for publication, for the following reasons.

We thank the reviewer for recognizing the general interest in the biological question we asked and the model we proposed.

1. The simulations are done with a purely 1D model for loop extrusion, and synapsis formation is essentially a first-passage time problem. While interesting I think this is a significant limitation, and it would be much more convincing to study at least some cases with 3D polymer simulation models where the polymer dynamics is considered (at least to prove when polymer dynamics is not relevant). This is important, I feel, because there are many biophysical details which are included in the modeling (targeted loading, different cohesin populations, LEF stabilisation) and which is not clear to me should be any more important than 3D effects such as polymer end diffusion, polymer concentration and stickiness for instance of DSB for each other or for the rest of the polymer. I understand it is not practical or required for this first contribution to consider 3D effects systematically, but at least the basic case should be done with full 3D polymer dynamics to make the study more convincing.

We thank the reviewer for the suggestion to incorporate 3D polymer simulation models. We added 3D polymer simulations with the following 3 parameter sets:

1) Passive diffusion alone (**Figure 1D**). We show that passive diffusion alone is too slow to achieve the synapsis kinetics observed in vivo.

2) Experimentally constrained loop extrusion parameters from the *Fbn2* locus (**Figure 2**) derived from a recent study from our group (Gabriele, Brandão, and Grosse-Holz et al *Science* 2022). We show this simple loop extrusion model modestly improves synapsis efficiency compared with passive diffusion alone, but cannot yet match the fast synapsis kinetics determined in vivo.

3) Parameter combination identified from 1D 5-dimension parameter scan that gave near-perfect synapsis efficiency (**Figure 5D**). We show this parameter set, in 3D simulations, achieves synapsis kinetics and efficiency consistent with *in vivo* observations.

In summary, we agree with the reviewer that 3D simulations were necessary to properly compare with 3D diffusion, and we believe these new simulations have greatly strengthened the manuscript.

2. The presentation of the results is difficult to follow, partially because all results, both positive and negative, are discussed in turn. I think the discussion of the subsequent mechanistic model improvement/modifications should be ideally more concise: a brief discussion of the biophysical reason for failure/partial success is perhaps OK, but too much technical discussion here is not so helpful in my view. The analytical theory is nice, but again can be presented more concisely I feel. The current presentation is more suitable for a more specialised journal (Biophys. J. for instance). The videos are I think more clear. They are very useful however they give me an overall impression that there are quite a few parameters which need to be tuned to get the desired effect.

We appreciate the suggestions and we have greatly shortened the presentation both of the various mechanistic modifications as well as the analytical theory. We believe the revised manuscript concisely conveys the most important points.

As minor comments:

1. The discussion of criteria for synapsis failure/formation given in the method section (page 18, Monitoring of synapsis events) is not too clear. Given their key relevance to the results, I think these criteria should be briefly clarified in the main text and a diagram here would help (along the lines of the modelling videos which are clearer).

We thank the reviewer for the suggestion of clarifying the criteria of synapsis failure/formation. We added **Supplementary Fig. 4** and a brief discussion to better illustrate the criteria for successful/failed synapsis.

2. I think it would be good to clarify from the beginning that the main model is purely 1D.

We thank the reviewer for this suggestion. With the incorporation of 3D polymer simulations, we added clarifications on which simulations were done in 3D, and which were done in 1D.

3. Did the authors consider what would happen for diffusing LEFs (e.g., Brackley et al., PRL 119, 138101 (2017))? This may be a useful/interesting case to consider to show whether active extrusion is crucial. Targeted loading of diffusing LEFs also leads to their ratcheting which may help diffusing LE as a potential mechanism.

We thank the reviewer for the suggestion. Given the near-infinite number of possible model extensions, and the reviewer's suggestion to keep the presentation concise, we do not explore this in our modeling, but we now cite the referenced work and note diffusing LEF could be an alternative model when discussing the limitations of our study (page 15 line 409): "... For example, we do not consider alternative extensions of loop extrusion based on diffusive sliding of LEFs (Brackley et al *Physical Review Letters* 2017)."

4. In reality there are foci formation at sites of repair which I believe include sites of DSBs. There are chromatin-binding proteins which bridge the DNA and cause microphase separation of heterochromatin and euchromatin and they are likely to have a strong effect on foci formation. Likewise, I think the effect of bridging proteins (e.g., HP1 or other multivalent transcription factors) on synapsis formation is potentially important and should be discussed as an alternative mechanism to look for, beyond what is considered by current model. I feel this is particularly relevant for active or silenced chromatin, less so for inert chromatin, for which this LE-mediated mechanism may be more relevant.

We thank the reviewer for these suggestions, which we have incorporated into the revised manuscript.

We discussed DDR foci and chromatin binding proteins driving microphase separation as alternative mechanisms to look for (page 15 line 412): "We have not considered the effect of DNA-damage response (DDR) foci (Polo et al *Genes & Development* 2011, Francia et al *Nature* 2012, Rossiello et al *Curr. Opin. Genet. Dev.* 2014, Fumagalli et al *PloS One* 2014, Willers et al *Seminars in Radiation Oncology* 2015, Clouaire et al *DNA Repair* 2017, Arnould et al *Nature* 2021) on synapsis. Repair factors recruited to the DDR foci such as

53BP1 could drive microphase separation to help constrain the DSB ends and promote synapsis (Dimitrova *Nature* 2008, Difilippantonio *Nature* 2008, Zhang et al *Nature Communications* 2022).” We also note that while foci/condensates formation may help, there is still a search problem. If the focus/condensate is small, it may help reduce the end-to-end distance required for synapsis, but the DSB ends still need to come sufficiently close. If the focus/condensate is large, the DSB ends still need to somehow find each other within the focus/condensate.

We added a discussion of the potential effect of HP1 and chromatin states (page 15 line 415): “The shift to a repressive chromatin state at the DSB through the accumulation of repressive complex including HP1, might help constrain the two DSB ends to facilitate synapsis (Gursoy-Yuzugullu et al *Journal of Molecular Biology* 2016).”

We also pointed readers to multivalent transcription factors for alternative mechanisms that might help constrain DSB ends (page 15 line 416): “Other chromatin-binding proteins such as multivalent transcription factors could also constrain DSB end movement by bridging two sides of a DSB (Izhar et al *Cell Reports* 2015).”

Finally, we note that our focus here is narrowly focused on “can loop extrusion help mediate DSB synapsis”. The number of additional mechanisms that may be operational at any given locus far exceeds what we can model (or have reasonably bound parameter estimates for), as such we prefer to note these as limitations of our current model and leave their detailed exploration for future work.

5. While there is a nice discussion of the potential effect of knockout/degrons on future experiments, it would be good to provide some more quantitative comparison with existing experimental data on DSB. For instance ChIP-seq and Hi-C are discussed in the Supplementary Information but it would be good to have a metric for agreement with experiments and an idea of how much each experiment constrains parameter values in the model.

We thank the reviewer for the suggestion of drawing quantitative comparisons with experimental data. For the fold enrichment of LEFs in DSB containing TADs, we added “% experimental enrichment” as a quantitative metric and found the simulated LEF enrichment with 250X, 500X, and 1000X targeted loading corresponded to 103%, 136% and 166% of the experimentally observed enrichment at steady state, respectively (**Supplementary Fig. 6A** pink text). For the Hi-C around DSB, we added “% experimental stripe pattern size” as a

quantitative metric and found the size of the simulated stripe pattern corresponded to ~ 109% experimentally observed stripe pattern size at steady state (**Supplementary Fig. 7A** pink text). We view these agreements as reasonably good given the complexity of the experimental systems.

Response to Reviewer 2

Yang, Brandao and Hansen constructed a mathematical model to test the intuitive hypothesis that loop extrusion may aid in synapsis of double-strand breaks. The paper is well written and guides the reader through the thought-process step by step. The figures and movies are clear and very helpful in explaining the research. In short, the form factor and style are excellent, and the authors provide a detailed explanation of their model in the Supplemental Information for transparency.

My main assessment of this manuscript relates to the assumptions the authors have used as the basis for their model, and the relevance of this study to previous work on DSB repair and synapsis.

Overall, the study presents interesting elements that might provide some modeling framework for DSB repair, but I find that many of the underlining assumptions and literatures that the authors used to build their model are limited, which results in a confirmatory model that provides a reductive view of DSB repair. Because the current work is lacking critical elements that are required for relating any DSB repair model to established experimental work I find that the study in its current state is not appropriate for publication. The main points are described below, and would require the authors to preform complete revisions of their model and manuscript.

We thank the reviewer for recognizing the quality of the presentation of our study. We also appreciate the suggestion of evaluating the assumptions we made in a broader context of prior experimental work, which we addressed in the revised manuscript.

Major points:

1. The authors sought to address the synapsis step of DSB repair via NHEJ (and also alternative NHEJ), but have omitted a large body of literature, while at the same time their studies rely only on references that seems to support their specific narrative and model. This sort of selectivity undermines the validity of their key assumptions, conclusions and thereby the entire premise of this study. In the introduction, and throughout the manuscript, the authors state that it is generally assumed that synapsis occurs via passive 3D diffusion, referencing a

review paper on NHEJ repair (Zao et al). This review provides a comprehensive survey on the molecular drivers of NHEJ repair, but contains little to no information on models for diffusion and synapsis.

The authors then claim that 3D diffusion is not efficient enough to promote DSB end synapsis in mammalian cells, to support this they use their model with parameter derived from modeling data of a single study (Taleei et al, 2012), but provide no justification of basing the entire model on a single study, while neglecting to mention many relevant studies where repair dynamics have been measured and modeled (for example, Reynold et al NAR 2012, Yue Liu et al JCS 2019). There are several other studies where the rate of repair by NHEJ of different DSBs have been measured or estimated by various methods. There are also multiple biochemical studies that examined the rate of repair products and derived the kinetics of synapsis. None of these results are being considered in the model, when the authors conclude that diffusion cannot facilitate efficient synapsis.

We apologize for the oversight of not citing all relevant literature. We have incorporated a more comprehensive survey of the relevant papers into the revised manuscript.

For the claim that synapsis is generally assumed to occur via passive 3D diffusion, we agree with the reviewer that the previously cited review paper (Zhao et al *Nature Reviews Molecular Cell Biology* 2020) is not the most suited reference here, and we have replaced that citation with six papers more directly supporting the claim (page 2 line 38): “Synapsis, the bringing of DSB ends back together for NHEJ, is generally assumed to be mediated by passive 3D diffusion (Kruhlak et al *Journal of Cell Biology* 2006, Soutoglou et al *Nature Cell Biology* 2007, Jakob et al *PNAS* 2009, Lucas et al *Cell* 2014, Graham et al *Molecular cell* 2016, Zhao et al *Nature communications* 2019)”.

We also agree that we should have better substantiated our claim that 3D diffusion is not efficient enough for DSB synapsis. In fact, Reviewer #1 also made this point. To demonstrate that 3D diffusion alone is inefficient to achieve fast synapsis kinetics observed *in vivo*, we have now performed 3D polymer simulations (**Fig. 1D**) and updated experimental estimates of *in vivo* synapsis kinetics by referencing several papers that most directly measured/estimated the rate of synapsis (page 3 line 67): “The most direct measurement of *in vivo* synapsis kinetics thus far, performed human osteosarcoma cells (U2OS), shows ~95% of DSBs are synapsed within 20 minutes (Reid et al *PNAS* 2015). Consistently, the average synapsis time is estimated to be around 6-17 minutes in mammalian cells based on prior experimental data (Reynolds et al *NAR* 2012, Taleei & Nikjoo *International Journal of Radiation Biology* 2012, Taleei &

Nikjoo *Mutation Research/Genetic Toxicology and Environmental Mutagenesis* 2013, Reid et al *PNAS* 2015).” We have also cited Yue Liu et al *JCS* 2019 for estimating Ku70/80 kinetics (page 19 line 540): “We motivated the 1 minute time scale as the mean expected time for the recruitment of DSB repair factors such as Ku70/80 which are necessary to facilitate synapsis (Reynolds et al *NAR* 2012, Liu et al *JCS* 2019, Zhao et al *Nature Reviews Molecular Cell Biology* 2020, Aleksandrov et al *Molecular Cell* 2018).”

We apologize for these oversights and hope the revised version satisfies the reviewer.

2. The authors’ main motivation for proposing and modeling that synapsis is driven by loop-extrusion is their conclusion that diffusion alone cannot be accounted for the known rates of productive synapsis and repair. They made this conclusion by calculating the encounter of the two sides of the DSB under 3D diffusion, while assuming that the reactive elements that promote synapsis are only localized at broken DNA ends. This assumption is very reductive and contrast many studies that demonstrated that the DNA ends serve as an initiating point for signaling and the recruitment of extended protein networks and formation of a repair foci. DSB induction leads to rapid changes to chromatin around the break and recruitment of a large protein complexes, such as 53BP1, driving phase separation to form condensates thereby extending the regions of reactivity well beyond the broken DNA ends. It is likely that 53BP1 can drive synapsis of the two ends enclosed within the foci, as it was shown to promote synapsis of neighboring and distal DSBs (Difilippantonio et al *Nature* 2008, Dimitrova et al *Nature* 2008).

We thank the reviewer for this suggestion. Another important motivation we highlighted for proposing the model of loop extrusion-mediated synapsis is the fact that recruitment of all reactive elements takes time, during which the two DSB ends may diffuse far apart (as was previously observed by Soutoglou *Nat Cell Bio* 2007). Loop extrusion could thus serve as a **pre-emptive** mechanism that prevents the two ends from diffusing far apart.

We agree with the reviewer that other elements not localized at the DSB ends such as DNA-damage response (DDR) foci might play a role in the synapsis process. We added a discussion on how the DDR foci might benefit synapsis and cited the suggested papers when discussing the limitations of our study (page 15 line 412): “We have not considered the effect of DNA-damage response (DDR) foci (Polo et al *Genes & Development* 2011, Francia et al *Nature* 2012, Rossiello et al *Curr. Opin. Genet. Dev.* 2014, Fumagalli et al *PLoS One* 2014, Willers et al *Seminars in Radiation Oncology* 2015, Clouaire et al

DNA Repair 2017, Arnould et al *Nature* 2021) on synapsis. Repair factors recruited to the DDR foci such as 53BP1 could drive microphase separation to help constrain the DSB ends and promote synapsis (Dimitrova *Nature* 2008, Difilippantonio *Nature* 2008, Zhang et al *Nature Communications* 2022).” We also note that while foci/condensates formation may help, there is still a search problem. If the focus/condensate is small, it may help reduce the end-to-end distance required for synapsis, but the DSB ends still need to come sufficiently close. If the focus/condensate is large, the DSB ends still need to somehow find each other within the focus/condensate.

A challenge with attempting to model many of these other mechanisms is that we do not currently have sufficient knowledge of the kinetic parameters for most of these mechanisms. In contrast, over the last few years, our kinetic and quantitative knowledge of loop extrusion dynamics has greatly improved and we can therefore model loop extrusion with higher confidence. As such, we feel it is best to name these other mechanisms as limitations of our current model and as interesting extensions for future work.

3. Beyond the properties of 53BP1 and the repair foci, the authors fail to include or even consider other key proteins of the NHEJ pathways that have no enzymatic role, and were shown to serve as scaffolding proteins that stimulate repair by enhancing synapsis. In particular, the authors would benefit from familiarizing themselves with many relevant studies showing that XLF, XRCC4 and LIG4 form extended structures (Hammel et al, *Structures* 2010, Ropars et al *PNAS* 2011, Andres et al *NAR* 2012, Brandi et al *Biochem Cell Biol* 2013) and that these structures can support dynamic synapsis away from the ends (Reid et al *PNAS* 2015, Brouwer et al *Nature* 2016). Other studies also showed that such synapsis can later efficiently transition to ligation products (Reid et al *NAR* 2017, Conlin et al *Cell Reports* 2017, Nemoz et al *NSMB* 2018). Importantly, it was demonstrated that these synapsis structures form at DSBs sites in cells. Overall, several studies, including some that are referenced above have proposed that the elongated structures that form in synapsis can enhance the rate of repair by reducing the dimensionality of the system and diffusion of the ends (Andres et al *NAR* 2012, Brandi et al *Biochem Cell Biol* 2013, Reid et al *PNAS* 2015). This concept was recently described for bacterial DNA repair (Wiktor et al *Nature* 2021) and likely presents an appropriate landscape for modeling many of the kinetics that occur in biological systems.

We thank the reviewer for pointing us in this direction. We agree that the long filament assembled around DSB ends by XLF, XRCC4 and LIG4 could facilitate dynamic synapsis and potentially allow the bridging of two DSB ends at a larger

distance. We added a discussion of this point and cited the suggested papers (page 15 line 418): “The long protein filaments formed around DSB ends by downstream NHEJ factors XRCC4, XLF and DNA ligase IV (Hammel et al *Structure* 2010, Ropars et al *PNAS* 2011, Andres et al *NAR* 2012, Brandi Mahaney et al *Biochemistry and Cell Biology* 2013) could also facilitate dynamic synapsis (Reid et al *PNAS* 2015, Brouwer et al *Nature* 2016), potentially by reducing the dimensionality of the system and allowing bridging of DSB ends at a larger distance (Andres et al *NAR* 2012, Brandi Mahaney et al *Biochemistry and Cell Biology* 2013, Reid et al *PNAS* 2015). Such dynamic synapsis structures at DSB sites could efficiently transition to NHEJ ligation products (Reid et al *PNAS* 2015, Conlin et al *Cell Reports* 2017, Nemoz et al *NSMB* 2018).”

We now explicitly note that a limitation of our model is that we have not yet modeled these mechanisms. We also note that the “reduced dimensionality” concept has so far been applied to HDR (Wiktor *et al. Nature* 2021), whereas we exclusively focus on NHEJ in G1. Extending our model to incorporate elongated filaments is an exciting future direction, though it will require detailed quantitative knowledge of all the biophysical parameters necessary to model it. As such we explicitly note the limitation of not considering it here, and leave it for future work.

4. The authors only consider a scenario whereby a synaptic complex form with the DNA-PK holoenzyme, but such a complex likely occurs as a final step prior to transition to the ligation step. In particular, it was shown that DNA-PKcs is dispensable for synapsis (Brouwer et al *Nature* 2016, Zhao et al *Nature Comm* 2019) and its specific roles during cellular repair might occur well after synapsis (Reynold et al *NAR* 2012, Yue Liu et al *JCS* 2019).

We thank the reviewer for pointing this out. We previously used the autophosphorylation of DNA-PKcs as a reference point to estimate the average synapsis time. We agree with the reviewer that this might lead to an overestimation of the average synapsis time given DNA-PKcs’ role in NHEJ. In our revised manuscript, following the reviewer’s suggestions in point 1, we have now added additional papers that do not rely on using DNA-PKcs as a reference point and thus provide a more general estimate of the average synapsis time (page 3 line 68): “... the average synapsis time is estimated to be around 6-17 minutes in mammalian cells based on prior experimental data (Reynolds et al *NAR* 2012, Taleei & Nikjoo *International Journal of Radiation Biology* 2012, Taleei & Nikjoo *Mutation Research/Genetic Toxicology and Environmental*

Mutagenesis 2013, Reid et al *PNAS* 2015).” Note that this range is slightly expanded compared with the range of 6-11 minutes in the original manuscript.

5. The authors should consider several additional active mechanisms that might contribute to DSB mobility, synapsis and repair, such as the involvement of nuclear actin and microtubules that were shown to affect repair (Lottersberger et al *Cell* 2015, Schrank BR *Nature* 2018, Zhu et al *Elife* 2020).

We thank the reviewer for the suggestion of considering the enhanced mobility of chromatin around DSBs mediated by nuclear actin and microtubules. We added a brief discussion on this (page 15 line 421): “Enhanced mobility of chromatin around DSBs mediated by nuclear actin and microtubules has been hypothesized to promote synapsis (Lottersberger et al *Cell* 2015, Zhu et al *Elife* 2020).” Schrank BR *Nature* 2018 focuses on HDR and we therefore decided to leave it out at this time. Since we are not aware of sufficient quantitative kinetic parameters to model nuclear actin and microtubules in synapsis and since this is beyond the scope of our current work, we instead note these as a limitation of our current model in the Discussion.

6. In modeling of loop extrusion rates, the authors consider several studies where the rates were measured in-vitro using naked DNA. These rates would be considerably different for chromosomal DNA in cells, and will likely be affected by the location of the DSB, and by the formation of DSB foci and recruitment of other chromatin factors. The authors should consider how to address these factors in their models.

We appreciate this suggestion. We agree with the reviewer that *in vivo* extrusion rate may differ from *in vitro* measurements. However, the absence of clear data on *in vivo* extrusion speeds still means that the *in vitro* measurements are currently the least bad estimates in our opinion. Arnould et al (*Nature* 2021) noted the speed of *in vivo* cohesin-mediated γ H2AX spreading is consistent with the \sim 1kb/s cohesin extrusion speed estimated from *in vitro* studies. Liu et al. (Liu *Science* 2020 DOI: 10.1126/science.aay820) proposed that the speed could be more than 100kb/minute, while we previously estimated a lower cohesin extrusion speed of \sim 0.2-0.25 kb/s at the *Fbn2* locus (Gabriele, Brandão, and Grosse-Holz et al *Science* 2022).

To make this uncertainty clear, we explicitly discuss this point: (page 15 line 423): “... we note that the *in vivo* extrusion speed is associated with large uncertainty and was estimated to be \sim 0.2-0.25 kb/s for cohesin at the *Fbn2* locus in mESCs (Gabriele, Brandão, and Grosse-Holz et al *Science* 2022). Furthermore, the extrusion speed may vary with different genomic loci, cell

types, and LEF types, and may change upon the recruitment of DSB repair factors.”

Additional points:

1. The author studied the LEF separation from 62.5 kb to 500 kb and found better synapsis efficiency with the lowest separation. Later, a separation of 125 kb was used for further simulation. The reason why 125 kb was chosen for further simulation is not apparent.

We thank the reviewer for pointing this out. We have now added the motivation of choosing separation = 125 kb (page 7 line 181): “Given the large number of processivity-separation combinations, hereafter we fix the LEF separation at 125 kb unless otherwise specified, which gives the highest genome coverage by LEFs among separation values estimated by studies comparing polymer simulations and experimental data (Fudenberg et al *Cell Reports* 2016, Fudenberg et al *Cold Spring Harbor Symposia on Quantitative Biology* 2017, Gassler et al *The EMBO Journal* 2017, Nuebler et al *PNAS* 2018, Banigan & Mirny *Elife* 2020, Banigan et al *Elife* 2020, Gabriele, Brandão, and Grosse-Holz et al *Science* 2022, Dequeker et al *Nature* 2022).”

2. The authors need to provide a physical basis as to why LEFs may be concentrated near the DSB sites in G1 phase.

We have now added the physical basis (by the MRN/MRX complex) for why LEFs may be concentrated near the DSB sites in the G1 phase when first introducing targeted loading of LEFs at DSBs (page 10 line 297): “...accumulation of cohesin subunit SCC1 at DSBs sites leading to a ~2-10-fold cohesin enrichment at restriction-enzyme induced DSB sites (Caron et al *PLoS Genetics* 2012, Cheblal et al *Molecular Cell* 2020, Arnould et al *Nature* 2021) possibly mediated by the MRN/MRX (MRN in mammalian cells, MRX in *Saccharomyces cerevisiae*) complex (Cheblal et al *Molecular Cell* 2020)...”

3. In figure 4B, does the yellow hexagon represent acetylation? It should be noted in the figure or the legend.

We have now noted in **Fig.4D** that the yellow hexagon represents acetylation.

4. Figure 4C / line 273: the authors claim that DSB stabilization leads to mildly improved synapsis. Given the position of the data points in Figure 4C, this interpretation seems to be a stretch. The data points basically form a flat line.

We have now added an inset in **Fig.4C** showing there is a small but statistically significant improvement in synopsis efficiency with LEF stabilization by DSB ends.

5. The authors point out recent studies that show LEFs can bypass each other. This undercuts one of the basic assumptions made in this paper.

In the revised manuscript we emphasized that the bypassing LEFs are a different **type** of LEFs (e.g. condensin), as opposed to the non-bypassing LEFs considered in our model (e.g. cohesin) on page 15 line 410: "Further extensions to our model could be explored by adding a different type of LEFs that could bypass each other, as supported by experiments showing condensin is able to bypass each other (Kim et al *Nature* 2020, Brandão et al *NSMB* 2021), which would help improve genome coverage by DNA loops."

6. Line 183: I am not sure if 28 is the correct reference.

We apologize for the confusion: 28 is an equation, we have now added "Eq." at the front.

7. When I downloaded the zipped software from Github, I was unable to extract it completely due to paths that were too long. The problem seems to be some files in the Data folder. I hope the authors can fix this.

We thank the reviewer for noticing this and apologize for the issue. We have fixed this issue.

Response to Reviewer 3

In this manuscript, Yang and coworkers present a theoretical study of DNA double-strand break (DSB)end synopsis and how loop extrusion would influence the synopsis efficiency and synopsis time.

The authors argue that passive thermal diffusion is not sufficient to bring back the two ends together to rejoin when a double-strand break occurs. They suggest that loop extrusion could be a way for the ends to come together within the biological timescales. They first examine a simple loop extrusion model and find that the efficiency is unsatisfactory. Then they add certain additional features into the model and argue that a combination of these factors would make the synopsis via extrusion highly efficient. They do simulation and also provide analytical arguments to support their claim.

This is indeed an interesting work. The theory and simulations are indeed valid. However, there are some concerns that need to be addressed to make it realistic and publishable in Nature Communications. The concerns are:

1) The authors assume that loop extrusion or loop associated proteins (boundary elements like CTCF, LEFs/cohesin) are the only players in this process. However, we know that there are many other proteins and interactions that hold the chromatin together.

We thank the reviewer for their feedback and the suggestion of considering other proteins and interactions involved in the synapsis process, which we addressed in the revised manuscript by discussing the potential role of DDR foci, microphase separation driven by 53BP1, multivalent transcription factors, long filaments formed by XRCC4/XLF/DNA ligase IV, nuclear actin and microtubules (page 15 line 412 to line 422). As our kinetic knowledge of these processes improves, it will be crucial to extend our model to incorporate these mechanisms in the future.

1a) We know that inter-nucleosome interactions do happen, and they can hold chromatin regions together, preventing them from diffusing away.

1b) We know that proteins like HP1, Polycomb complexes, etc., hold chromatin together.

1c) Even in the active regions, proteins like FOXA1 etc bind between different chromatin segments, effectively preventing them from diffusing away.

We thank the reviewer for suggestions 1a)-1c).

For 1a), we would like to distinguish between “preventing the ends from diffusing away right after DSBs occurrence” and “preventing the ends from diffusing away during the synapsis process”. For the former, live-imaging experiments (Soutoglou et al *Nat Cell Biol* 2007) shows that the distance between the two ends at most DSBs has an initial increase of ~120 nm increasing up to a mean distance of ~220 nm after DSB induction (Fig. 2e in Soutoglou et al *Nat Cell Biol* 2007) and in some cases the distance reaches well above 300 nm (Fig. 2d in Soutoglou et al *Nat Cell Biol* 2007). These distances are much greater than the diameters of nucleosome/HP1/Polycomb/FOXA1 (~10 nm), suggesting their role in holding the two ends together right after DSB occurrence is limited if any. However, we do agree that these forces could potentially help constrain the two DSB ends during synapsis, and we found experimental evidence for the proteins suggested in 1b) and 1c).

In response to 1b), we added a brief discussion on how the accumulation of HP1 at DSB might help constrain the two DSB ends (page 15 line 415): “The shift to a repressive chromatin state at the DSB through the accumulation of repressive complex including HP1, might help constrain the two DSB ends to facilitate synapsis (Gursoy-Yuzugullu et al *Journal of Molecular Biology* 2016).”

In response to 1c), we discussed in the revised manuscript the potential role of multivalent transcription factors in constraining DSB end movement (page 15 line 416): “Other chromatin-binding proteins such as multivalent transcription factors could also constrain DSB end movement by bridging two sides of a DSB (Izhar et al *Cell Reports* 2015).” Thus, we now explicitly acknowledge that it is a limitation of our current model to not directly model these.

1d) Equivalently, there is also the micro "phase separation" phenomenon in parallel with loop extrusion; wouldn't that also be important in deciding the efficiency?

We appreciate this suggestion. We added a discussion on how microphase separation could help constrain DSB ends to promote synapsis (page 15 line 413): “Repair factors recruited to the DDR foci such as 53BP1 could drive microphase separation to help constrain the DSB ends and promote synapsis (Dimitrova *Nature* 2008, Difilippantonio *Nature* 2008, Zhang et al *Nature Communications* 2022).”

Why would an estimate without accounting for any of the factors above be accurate?

This is a fair point. If we desire maximal accuracy, no general model will be possible. For example, we already know that DSB synapsis kinetics will differ greatly between e.g. heterochromatin and euchromatin, during S-phase and during G1, between different cell types (e.g. postmitotic neuron vs. stem cell), between aged cells and totipotent cells, will depend on metabolism and possible circadian rhythms, epigenetics, and so forth. Thus, a 100% accurate model is in our view impossible. Furthermore, if we were to model all known possible mechanisms, the number of parameters would be so high that it would be impossible to meaningfully constrain them.

However, our goal is different. Our goal is to specifically compare “3D diffusion only” vs. “3D diffusion + loop extrusion” to assess whether loop extrusion is likely to contribute to synapsis. Thus, though our model is quantitative, our goal is more conceptual and mechanistic - to show that loop extrusion likely plays a role in mediating DSB synapsis, which we now emphasize in the revised

manuscript (page 15 line 428): “In summary, a limitation of our study is that it does not account for all possible mechanisms. Instead, our focus here is to specifically evaluate if and how loop extrusion can mediate synapsis and contribute above and beyond passive 3D diffusion. Nevertheless, as our detailed quantitative knowledge of these additional mechanisms improves, it will be important to extend our model to incorporate these additional mechanisms in the future..”

2) The authors say that their one bead is 1kb of chromatin. The relevant length scale for DSB and its resolution is 1bp-10bp. In 1kb, there are about 5 nucleosomes, and it is a large region (compared to the bp length-scale relevant for DSB). As mentioned in (1a) above, inter-nucleosomal interaction, etc, would already play a role within that 1kb "bead". So why would a model without basepair resolution be relevant for studying DSB?

We thank the reviewer for pointing this out. We chose a 1kb monomer size for the following two reasons: (1) an appropriate level of coarse-graining is necessary (even with 1 kb beads, it took many month to finish the 3D polymer simulations included in the revisions); (2) we noted in the revised manuscript (page 7 line 205) that when the two ends are sufficiently close (when the unextruded DNA within constraining LEF is smaller than ~2kb), synapsis by passive diffusion is highly efficient (synapsis time on the order of seconds), as supported by experimental observation (Graham et al *Molecular Cell* 2016) and theoretical work (Amitai & Holcman *Physical review letters* 2013).

Taken together, once the synapsis criteria are met under our coarse-grained model, the additional time taken to achieve basepair resolution matching of the two DSB ends is on the order of seconds, which is negligible compared with the synapsis time on the order of minutes predicted by our model (also consistent with experimental observations). Therefore we circumvented the need for a basepair resolution model, which would be computationally infeasible.

2a) The authors take 2million+ lattice sites with 1kb length suggesting a total length of 2billion+ bp. What is the relevance? The longest human chromosome is of the order of 250 Mbps, right?. How are these numbers relevant?

We thank the reviewer for this question. Given a processivity of around 250 kb for cohesin, for our loop extrusion simulations the relevant length scale is hundreds of kb, which is why it should not matter to our results whether we use chromosomes of tens of megabases or gigabases.

As a robustness test, we performed simulations using either 1) one 2165 Mb chromosome or 2) fifteen 144 Mb chromosomes (average human chromosome length 3.3Gb/23~144 Mb). As shown from the simulation below, our results are identical within error. We conclude that the choice of chromosome length does not affect our findings.

3) It is a limitation that the events are happening in 3D, and the model in this manuscript is a 1D model. It is acceptable as a physics model for loop extrusion estimates. Would the answer be comparable if a similar work is done with a 3D model?

Yes, this is a really good point and we agree that explicitly modeling 3D diffusion is necessary. In fact, Reviewer #1 and #2 also raised this point. 3D polymer simulations are very computationally expensive but we have now added 3D polymer simulations with the following 3 parameter sets:

1) Passive diffusion alone (**Figure 1D**). We show that passive diffusion alone is too slow to achieve the synapsis kinetics observed *in vivo*.

2) Experimentally constrained loop extrusion parameters from the *Fbn2* locus (**Figure 2**). We show this simple loop extrusion model improves synapsis efficiency compared with passive diffusion alone, but cannot yet match the fast synapsis kinetics determined *in vivo*.

3) Parameter combination identified from 1D 5-dimension parameter scan that gave near-perfect synapsis efficiency (**Figure 5D**). We show this parameter set, in 3D simulations, achieves synapsis kinetics and efficiency consistent with *in vivo* observations.

4) The 3D diffusion is taken to be of the order of 10^{-3} micron² per second based on experiments.

We thank the reviewer for pointing this out. The 3D polymer simulations we incorporated into the revised manuscript have a 3D diffusion coefficient of $0.7 \cdot 10^{-3}$ micron² per second.

4a) One would naively presume that the experimentally obtained diffusion coefficient value will already have the effect of loop extrusion, LEFs etc in it. Why would one need to add these separately?

We thank the reviewer for recognizing this subtlety. Since our original submission, our paper using super-resolution live-cell imaging of the *Fbn2* locus came out (Gabriele, Brandão, and Grosse-Holz et al *Science* 2022). This paper measures the diffusion coefficient both under WT conditions (including loop extrusion) and after acute RAD21 depletion (without loop extrusion). We use the diffusion coefficient without loop extrusion to avoid this confounding effect.

4b) There is a wide range of estimates for diffusion coefficient. For example, the Maeshima and other labs' experiments seem to suggest that the motion is subdiffusive, and the mean square displacement is about 10^{-2} micron² in 0.1 seconds (this would be like 0.1 micron² per second) [eg. Nozaki et al. and other other papers]

We thank the reviewer for pointing this out. The polymers in our 3D polymer simulations are also subdiffusive with $MSD \sim t^{0.5}$ (Gabriele, Brandão, and Grosse-Holz et al *Science* 2022), consistent with Maeshima and other labs' experiments. We agree with the reviewer that there is a wide range of estimates for the diffusion coefficient; however, the vast majority of estimates based on experiments are on the order of 10^{-3} micron² per second or lower, as nicely summarized in Miné-Hattab & Rothstein *Trends in Cell Biology* 2013, with which passive diffusion alone would be too slow to match synapsis kinetics observed *in vivo*. We now highlight the uncertainty in the chromatin diffusion coefficient and the recent estimates based on live cell imaging data in mammalian cells (page 18 line 535): "We note there is a wide range of estimates for the chromatin diffusion coefficient (Miné-Hattab & Rothstein *Trends in Cell Biology*

2013, Dion et al *Cell* 2013, Shinkai et al *PLoS Computational Biology* 2016), and the value used in this study is consistent with estimates on the order of $1 \cdot 10^{-3}$ $\mu\text{m}^2/\text{s}$ based on recent live-cell imaging experiments in mammalian cells (Tiana et al *Biophysical Journal* 2016, Gabriele, Brandão, and Grosse-Holz et al *Science* 2022).”

5) It is found that the boundary strength has little effect. It is not clear how BEs are placed (with orientation etc?). It is written that "We place BEs on the chromosome so that TADs of the sizes shown in Fig. 2A are achieved". However, it is not clear exactly how this is done. Is there an algorithm to place BES to achieve a certain TAD size?

We apologize for the confusion. In our revised manuscript we added clarifications for the placement of BEs and the choice of TAD sizes (page 17 line 489) “We placed pairs of BEs on the chromosome (each pairs consisting of two divergently oriented BEs) with the spacing between each pair ranging from 200 kb to 1200 kb (**Fig. 2A**), since TAD sizes range from 200 kb to 2.5 Mb based on experimental data (Gong et al *Nature Communications* 2018, Oluwadare & Cheng *BMC Bioinformatics* 2017).”

6) The underlying assumption is that all regions of the heterogeneous chromosome will have a similar distribution of LEFs/extrusion etc. However, things could be very different in heterochromatin and euchromatin regions. Is this result valid for both heterochromatin and euchromatin? DSB could happen anywhere, and the timescale could be very different in heterochromatin and euchromatin regions.

We thank the reviewer for this suggestion. In the revised manuscript, we highlighted that LEF distribution may not be uniform throughout the genome (page 15 line 426): “Finally, we have assumed a uniform distribution of LEFs throughout the genome, while previous work suggests that cohesin loading may not be uniform across the genome (Lengronne et al *Nature* 2004, Newkirk et al *Clinical epigenetics* 2017, Davidson & Jan-Michael *Nature Reviews Molecular Cell Biology* 2021).” We also pointed out that LEF extrusion speed might depend on genomic context (page 15 line 425): “... the extrusion speed may vary with different genomic loci ...”

We also note that our goal is to specifically compare “3D diffusion only” to “3D diffusion as well as loop extrusion” to assess whether loop extrusion is likely to contribute to synapsis. Thus, though our model is quantitative, our goal is more conceptual and mechanistic - to show that loop extrusion likely plays a role in mediating DSB synapsis, which we now emphasize in the revised manuscript

(page 15 line 428): “In the future and as our knowledge of these additional mechanisms improve, it will be important to extend our model to incorporate them to enhance our model's quantitative accuracy. We note that such limitations did not hinder us from addressing the conceptual and qualitative question we set forth--whether loop extrusion could add to diffusion to accelerate synapsis, as we demonstrated a qualitative leap in synapsis speed from the diffusion baseline achievable by loop extrusion.”

6) Why the iterative correction is applied to the simulated data? Typically iterative correction is to eliminate various biases and limitations in experiments, right? Everything, including normalization, is clear in a simulation, and there is no unknown information as far as I understand.

We thank the reviewer for this question. We performed ICE on the simulated data in the same way as the experimental data to allow a direct and fair comparison: ICE does not assume the source of biases, and some loci in our simulated data still have higher visibility than others as we only included LEFs that were within ± 2.5 Mb of the DSB sites (LEFs connecting loci within ± 2.5 Mb of the DSB sites and loci outside of ± 2.5 Mb were discarded to mimic experimental conditions), which was accounted for by ICE.

REVIEWERS' COMMENTS

Reviewer #3 (Remarks to the Author):

The revised manuscript and comments from authors address all my concerns. The manuscript may be published in Nature Communications.

Apart from an earlier review of my revision comments (referee-3), I have now reviewed responses to referee-1 as well. The authors have addressed all of the concerns. The authors have also addressed all the questions by reviewer 1. The major comment by referee-1 was to do a 3D polymer simulation. The authors have performed 3D simulations in the revised manuscript. That satisfactorily answers the first set of comments. The second comment was about the presentation of the paper. The presentation is suitable for Nature communications. The other comments were about discussing and adding appropriate citations in the context of phase separation etc. The authors have addressed all the remaining comments in the revised version. I recommend the publication of the manuscript

Reviewer #4 (Remarks to the Author):

The authors have properly addressed the concerns of Reviewer #2. They focused their study merely on the contributions of "loop extrusion" to the synopsis of DSB and appropriately discussed the limitations of the study including not considering other possible contributing factors, e.g. repair factors, as Reviewer #2 raised.

I have two additional points.

1. Could the authors have an estimation of how close the two DSB ends could be brought together by the LEFs? The synopsis by LEFs proposed in this manuscript is not specific to NHEJ and could apply to other DSB repair processes, such as alternative end-joining and SSA. The DSB ends would be eventually bridged by the repair factors (e.g. Ku70/80-X4L4-XLF/Pol θ /RAD52) to complete the repair. This has been confirmed by lots of studies showing ends synapsis by repair factors followed by direct repair. I assume the synapsis reported in this manuscript would not represent a close synapsis, namely, the two DSB ends are still apart from each other and could not be repaired by repair proteins. The authors should at least clarify the difference between the synapsis reported in this manuscript and the one used in the NHEJ community.

2. Some DSBs are tightly regulated, for example, DSB generated during V(D)J recombination. The DSB ends from V(D)J recombination might not diffuse away. These ends are regulated to transfer to the NHEJ factors. The loop extrusion mechanism might not apply to this scenario, instead, the loop extrusion mechanism might be applied for RAG1/2 scanning the RSS in the genome during V(D)J recombination, which is before the DSB formation. The authors might want to discuss this in the manuscript.

Response to Reviews

We thank all reviewers for their valuable comments, insights, and questions throughout the revision process. We have carefully reviewed and addressed all reviewers' comments. Overall, we believe that this revised manuscript is greatly strengthened and fully addresses the reviewers' comments.

Response to Reviewer #3

The revised manuscript and comments from authors address all my concerns. The manuscript may be published in Nature Communications.

Apart from an earlier review of my revision comments (referee-3), I have now reviewed responses to referee-1 as well. The authors have addressed all of the concerns. The authors have also addressed all the questions by reviewer 1. The major comment by referee-1 was to do a 3D polymer simulation. The authors have performed 3D simulations in the revised manuscript. That satisfactorily answers the first set of comments. The second comment was about the presentation of the paper. The presentation is suitable for Nature communications. The other comments were about discussing and adding appropriate citations in the context of phase separation etc. The authors have addressed all the remaining comments in the revised version. I recommend the publication of the manuscript

We thank the reviewer for recognizing that the revised manuscript has fully addressed the reviewers' comments.

Response to Reviewer #4

The authors have properly addressed the concerns of Reviewer #2. They focused their study merely on the contributions of "loop extrusion" to the synapsis of DSB and appropriately discussed the limitations of the study including not considering other possible contributing factors, e.g. repair factors, as Reviewer #2 raised.

I have two additional points.

1. Could the authors have an estimation of how close the two DSB ends could be brought together by the LEFs? The synapsis by LEFs proposed in this manuscript is not specific to NHEJ and could apply to other DSB repair processes, such as alternative end-joining and SSA. The DSB ends would be eventually bridged by the repair factors (e.g. Ku70/80-X4L4-XLF/Pol θ /RAD52) to complete the repair. This has been confirmed by lots of studies showing ends synapsis by repair factors followed by direct repair. I assume the synapsis reported in this manuscript would not represent a close synapsis, namely, the two DSB ends are still apart from each other and could not be repaired by repair proteins. The authors should at least clarify the difference between the synapsis reported in this manuscript and the one used in the NHEJ community.

We thank the reviewer for this suggestion. It is challenging to estimate how close the two DSB ends could be brought together by the LEFs, since diffusion of the two ends happens concurrently with the loop extrusion-mediated synapsis process, and when the two ends are sufficiently close (when the unextruded DNA within constraining LEF is smaller than ~2kb), diffusion-mediated synapsis happens within the order of seconds (Amitai & Holcman *Physical review letters* 2013; Graham et al *Molecular cell* 2016). In other words, before the theoretical distance limit between two DSB ends achievable by LEFs alone is reached, close synapsis would most likely have occurred with diffusion acting in parallel.

We agree with the reviewer that due to the coarse-graining necessary for having a reasonable speed of computation, the synapsis events recorded in our study do not precisely correspond to the close synapsis used in the NHEJ community. We have now added clarification in the main text (page 16 line 550): “Due to the coarse-graining necessary for efficient simulations we used a synapsis threshold of ~2 kb which is larger than the actual distance between the two DSB ends held together by the synaptic complex (Chen et al *Nature* 2021). However, when the length of unextruded DNA is on the order of 2 kb, diffusion of the two DSB ends is so efficient that the DSB ends will be aligned within seconds for downstream ligation (Amitai & Holcman *Physical review letters* 2013; Graham et al *Molecular cell* 2016), thus making a negligible contribution to the synapsis time.”

2. Some DSBs are tightly regulated, for example, DSB generated during V(D)J recombination. The DSB ends from V(D)J recombination might not diffuse away. These ends are regulated to transfer to the NHEJ factors. The loop extrusion mechanism might not apply to this scenario, instead, the loop extrusion mechanism might be applied for RAG1/2 scanning the RSS in the genome during V(D)J recombination, which is before the DSB formation. The authors might want to discuss this in the manuscript.

We appreciate the reviewer for pointing out that the role of loop extrusion likely precedes DSB in V(D)J recombination. We agree this is an important point and added a discussion on this in the main text (page 10 line 336): “It is worth noting that loop extrusion's role in V(D)J recombination likely precedes DSB occurrence (Zhang et al *Nature* 2019, Ba et al *Nature* 2020, Dai et al *Nature* 2021), since the DSB ends are already in proximity when DSB occurs and are kept close by the post-cleavage complex along and other DDR repair factors (Libri et al *Frontiers in Genetics* 2022).”